# OPENMIXUP: A COMPREHENSIVE MIXUP BENCHMARK FOR VISUAL CLASSIFICATION

## ABSTRACT

Data mixing, or mixup, is a data-dependent augmentation technique that has greatly enhanced the generalizability of modern deep neural networks. However, a full grasp of mixup methodology necessitates a top-down hierarchical understanding from systematic impartial evaluations and empirical analysis, both of which are currently lacking within the community. In this paper, we present OpenMixup, the *first* comprehensive mixup benchmarking study for supervised visual classification. OpenMixup offers a unified mixup-based model design and training framework, encompassing a wide collection of data mixing algorithms, a diverse range of widely-used backbones and modules, and a set of model analysis toolkits. To ensure fair and complete comparisons, large-scale standard evaluations of various mixup baselines are conducted across 12 diversified image datasets with meticulous confounders and tweaking powered by our modular and extensible codebase framework. Interesting observations and insights are derived through detailed empirical analysis of how mixup policies, network architectures, and dataset properties affect the mixup visual classification performance. We hope that OpenMixup can bolster the reproducibility of previously gained insights and facilitate a better understanding of mixup properties, thereby giving the community a kick-start for the development and evaluation of new mixup methods. The source code is publicly available.

## 1 INTRODUCTION

Large-scale deep neural networks (DNNs) trained with a large amount of annotated data have achieved remarkable success in visual representation learning (He et al., 2016; 2017). Nevertheless, DNNs with numerous parameters exhibit a significant generalization degradation evaluated on unseen samples when the training data is limited (Wan et al., 2013). To tackle this challenge, a series of data augmentation algorithms have been proposed and have progressively assumed a pivotal role in broadening data distribution diversity, thereby enhancing the robustness of DNNs and promoting visual recognition performance.

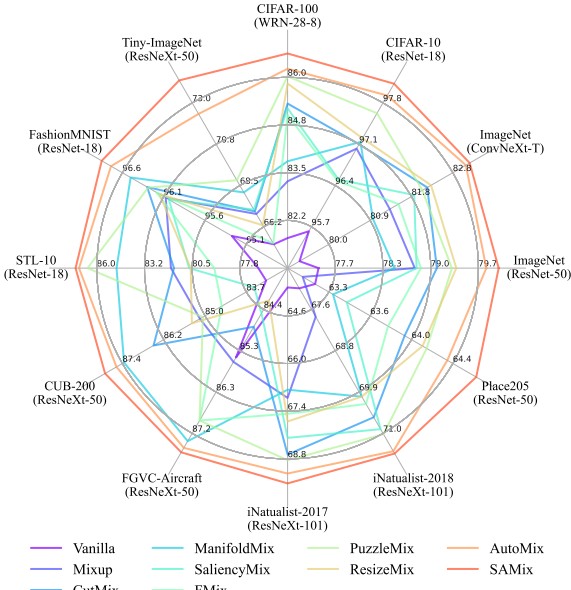

Figure 1: Top-1 accuracy radar plots of CNN-based mixup benchmark on 12 classification datasets.

Data mixing, which increases data diversity via mixing samples and their corresponding labels, has been demonstrated to be effective in improving the generalizability of DNNs, with notable success in visual classification tasks. Mixup (Zhang et al., 2018), as a pioneering work, generates virtual mixed samples with mixed labels through the convex combination of two input samples and the corresponding labels. ManifoldMix variants (Verma et al., 2019; Faramarzi et al., 2020) expand this operation to the hidden space. CutMix (Yun et al., 2019) introduced another type of sample mixing policy, which randomly cuts an input rectangular region and pastes it onto the target at the identical location. Subsequent works (Harris et al., 2020; ha Lee

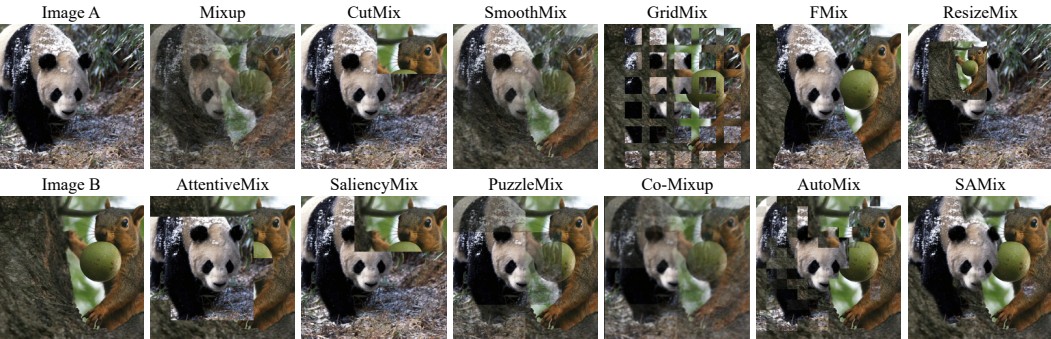

Figure 2: Visualization of mixed samples of supported mixup augmentation methods on ImageNet-1K. We choose the mixing ratio $\lambda = 0.5$ for a comprehensive comparison. Note that mixed samples are more precisely in *dynamic* mixing policies than these *static* ones.

et al., 2020; Baek et al., 2021) endeavored to mix the data more precisely in a *hand-crafted* way, which can be referred to as *static* mixing policies.

Despite the attempts to incorporate saliency information (Walawalkar et al., 2020; Uddin et al., 2020; Qin et al., 2023) into *static* mixing process, these methods still fall short in ensuring the inclusion of desired target objects within the mixed samples. As a result, this may give rise to the issue of label mismatch. To address this challenge, a series of optimization-based *dynamic* mixing methods are put forth, as illustrated in the second row of Figure 2, where offline optimizable algorithms are employed to refine the saliency regions and generate semantically richer virtual samples. With optimal transport, PuzzleMix (Kim et al., 2020) and Co-Mixup (Kim et al., 2021) present more enhanced policies in mixup mask determination. More recently, TransMix (Chen et al., 2022), TokenMix (Liu et al., 2022a), MixPro (Zhao et al., 2023), and SMMix (Chen et al., 2023) are specially designed for Vision Transformers (Dosovitskiy et al., 2021) (not for CNNs). Distinctively, AutoMix (Liu et al., 2022d) introduces a novel online training pipeline where the mixup generation is learned by a sub-network in an end-to-end manner, which generates mixed samples via feature maps and the mixing ratio.

**Why do we need a comprehensive mixup benchmark?** While *dynamic* mixing has almost demonstrated superior performance compared to the *static* one, its indirect optimization process incurs heavy computational overhead, which can greatly hinder the efficiency of data augmentation. Therefore, it is not advisable to naively transform the mixup policy from *static* interpolation to a *dynamic* manner without a top-down hierarchical understanding. Benchmarks play a pivotal role in driving advancements by establishing an agreed-upon set of tasks, impartial comparisons, and assessment criteria. To the best of our knowledge, however, there has been **no such comprehensive benchmarking study for the mixup community** to drive progress in a standard manner.

**Why do we need a standardized open-source framework?** Noticeably, most existing mixup algorithms are developed with diverse angles of enhancement, applying distinct configurations and coding styles **without a unified open-source coding framework** for streamlined pre-processing, method design, training, and evaluation. This lack of standardization poses obstacles to impartial evaluation and user-friendly development, necessitating substantial trial-and-error for practitioners. Therefore, it is much more crucial to first construct an all-in-one mixup development framework working across a range of scenarios instead of just coming up with new mixup strategies.

To this end, we introduce the *first* comprehensive mixup visual classification benchmark and a unified mixup-based model design and training framework called OpenMixup. Different from previous work (Naveed, 2021; Lewy & Madziuk, 2023), OpenMixup provides a standardized modular framework to conduct impartial comparisons and standard evaluations on mixup methods over a diversity of visual classification datasets. To achieve comprehensive and impartial comparisons, a series of mixup methods that represent the foremost strands of mixup visual classification is evaluated from scratch on 12 diversified datasets powered by the proposed framework, as shown in Figure 1, Figure 4. The results deliver valuable empirical insights incorporating diverse evaluation metrics as in Figure 6 and Figure A7, which can assist in discerning the elements leading to the method's success.

It is worth emphasizing that such a first-of-its benchmarking study is both time-consuming and resource-intensive, but we weighed it to be worthwhile for the community and carried it out from scratch. Interesting observations and insights are derived in Sec. 4 through extensive empirical analysis on how mixup properties and slight configuration changes influence downstream performance. In addition, although mixup exerts its power in semi-supervised (Berthelot et al., 2019; Liu et al., 2022c),

self-supervised (Kalantidis et al., 2020; Shen et al., 2022), and other downstream tasks (Wu et al., 2022; Bochkovskiy et al., 2020), it is derived from its supervised classification counterpart with shared attributes. Thus, we have limited the scope to include solely the most representative visual classification scenarios. Our contributions are summarized as three aspects:

- We present OpenMixup, the *first* comprehensive mixup visual classification benchmark, where 16 representative mixup algorithms are impartially evaluated from scratch across 12 visual classification datasets, ranging from classical iconic scenarios to fine-grained, long tail, and scenic cases. The datasets exhibit varying levels of granularity and scale, spanning from coarse-grained to fine-grained, encompassing both small-scale and large-scale datasets. This benchmark enables researchers to objectively evaluate their models against a broad set of baselines, providing a clear measurement of the method's effectiveness.
- We provide a unified model design and training codebase framework for customized mixup-based visual classification. The overall framework is built up with modular constituent elements (*e.g.*, data pre-processing, mixup algorithms, well-known backbones, optimization policies, and distributed training, *etc.*), which can enhance the accessibility and customizability of mixup-related computer vision downstream deployment and applications.
- Extensive empirical analyses are conducted on a wide scope of scenarios. Interesting observations are obtained with diverse analysis toolkits from our proposed framework, indicating significant potential for future methods to facilitate more systematic advancements.

## 2 BACKGROUND AND RELATED WORK

### 2.1 PROBLEM DEFINITION

**Mixup training.** We first consider the general image classification task with $k$ different classes: given a finite set of $n$ samples $X = [x_i]_{i=1}^n \in \mathbb{R}^{n \times W \times H \times C}$ and their ground-truth class labels $Y = [y_i]_{i=1}^n \in \mathbb{R}^{n \times k}$, encoded by a one-hot vector $y_i \in \mathbb{R}^k$. We seek the mapping from the data $x_i$ to its class label $y_i$ modeled by a deep neural network $f_\theta : x \longmapsto y$ with network parameters $\theta$ by optimizing a classification loss $\ell(.)$, say the cross entropy (CE) loss,

$$\ell_{CE}(f_\theta(x), y) = -y \log f_\theta(x). \tag{1}$$

Then we consider the mixup classification task: given a sample mixup function $h$, a label mixup function $g$, and a mixing ratio $\lambda$ sampled from $Beta(\alpha, \alpha)$ distribution, we can generate the mixup data $X_{mix}$ with $x_{mix} = h(x_i, x_j, \lambda)$ and the mixup label $Y_{mix}$ with $y_{mix} = g(y_i, y_j, \lambda)$, where $\alpha$ is the hyper-parameter. Similarly, we learn $f_\theta : x_{mix} \longmapsto y_{mix}$ by mixup cross-entropy (MCE) loss,

$$\ell_{MCE} = \lambda \ell_{CE}(f_\theta(x_{mix}), y_i) + (1 - \lambda) \ell_{CE}(f_\theta(x_{mix}), y_j). \tag{2}$$

**Mixup reformulation.** Comparing Eq. (1) and Eq. (2), the mixup training has the following features: (1) extra mixup policies, $g$ and $h$, are required to generate $X_{mix}$ and $Y_{mix}$. (2) the classification performance of $f_\theta$ depends on the generation policy of mixup. Naturally, we can split the mixup task into two complementary sub-tasks: (i) mixed sample generation and (ii) mixup classification (learning objective). Notice that the sub-task (i) is subordinate to (ii) because the final goal is to obtain a stronger classifier. Therefore, from this perspective, we regard the mixup generation as an auxiliary task for the classification task. Since $g$ is generally designed as a linear interpolation, i.e., $g(y_i, y_j, \lambda) = \lambda y_i + (1 - \lambda) y_j$, $h$ becomes the key function to determine the performance of the model. Generalizing previous offline methods, we define a parametric mixup policy $h_\phi$ as the sub-task with another set of parameters $\phi$. The final goal is to optimize $\ell_{MCE}$ given $\theta$ and $\phi$ as:

$$\min_{\theta, \phi} \ell_{MCE}\Big(f_\theta\big(h_\phi(x_i, x_j, \lambda)\big), g(y_i, y_j, \lambda)\Big). \tag{3}$$

### 2.2 SAMPLE MIXING

Within the realm of visual classification, prior research has primarily concentrated on refining the sample mixing strategies rather than the label mixing ones. In this context, most of the sample mixing methods are categorized into two groups: *static* policies and *dynamic* policies, as presented in Table 1.

**Static Policies.** The data mixing procedure in all *static* mixup policies is performed in a *hand-crafted* way. Mixup (Zhang et al., 2018) generates artificially mixed samples with mixed labels through the convex combination of two randomly chosen samples and the associated one-hot labels.

Table 1: Information of the supported visual Mixup augmentation methods in OpenMixup. Note that Mixup and CutMix in the label mixing policies indicate mixing labels of two samples by linear interpolation or calculating the cut squares.

| Method | Category | Publication | Sample Mixing | Label Mixing | Extra Cost | ViT only |
|---|---|---|---|---|---|---|
| Mixup (Zhang et al., 2018) | Static | ICLR'2018 | Hand-crafted Interpolation | Mixup | ✗ | ✗ |
| CutMix (Yun et al., 2019) | Static | ICCV'2019 | Hand-crafted Cutting | CutMix | ✗ | ✗ |
| SmoothMix (ha Lee et al., 2020) | Static | CVPRW'2020 | Hand-crafted Cutting | CutMix | ✗ | ✗ |
| GridMix (Baek et al., 2021) | Static | PR'2021 | Hand-crafted Cutting | CutMix | ✗ | ✗ |
| ResizeMix (Qin et al., 2023) | Static | CVMJ'2023 | Hand-crafted Cutting | CutMix | ✗ | ✗ |
| ManifoldMix (Verma et al., 2019) | Static | ICML'2019 | Latent-space Mixup | Mixup | ✗ | ✗ |
| FMix (Harris et al., 2020) | Static | arXiv'2020 | Fourier-guided Cutting | CutMix | ✗ | ✗ |
| AttentiveMix (Walawalkar et al., 2020) | Static | ICASSP'2020 | Pretraining-guided Cutting | CutMix | ✓ | ✗ |
| SaliencyMix (Uddin et al., 2020) | Static | ICLR'2021 | Saliency-guided Cutting | CutMix | ✗ | ✗ |
| PuzzleMix (Kim et al., 2020) | Dynamic | ICML'2020 | Optimal-transported Cutting | CutMix | ✓ | ✗ |
| AlignMix (Venkataramanan et al., 2022) | Dynamic | CVPR'2022 | Optimal-transported Interpolation | CutMix | ✓ | ✗ |
| AutoMix (Liu et al., 2022d) | Dynamic | ECCV'2022 | End-to-end-learned Cutting | CutMix | ✓ | ✗ |
| SAMix (Li et al., 2021) | Dynamic | arXiv'2021 | End-to-end-learned Cutting | CutMix | ✓ | ✗ |
| TransMix (Chen et al., 2022) | Dynamic | CVPR'2022 | CutMix+Mixup | Attention-guided | ✗ | ✓ |
| SMMix (Chen et al., 2023) | Dynamic | ICCV'2023 | CutMix+Mixup | Attention-guided | ✗ | ✓ |
| DecoupledMix (Liu et al., 2022c) | Static | arXiv'2022 | Any Sample Mixing Policies | DecoupledMix | ✗ | ✗ |

ManifoldMix (Verma et al., 2019; Faramarzi et al., 2020) variants extend the identical technique to latent spaces. CutMix (Yun et al., 2019) involves the random replacement of a certain rectangular area inside the input data while concurrently employing Dropout throughout the mixing process. Drawing inspiration from CutMix, several researchers have explored the use of saliency information (Uddin et al., 2020) to pilot mixing patches, while others have developed more advanced mixing strategies (Harris et al., 2020; Baek et al., 2021; ha Lee et al., 2020) with more complex *hand-crafted* design elements.

**Dynamic Policies.** In contrast to *static* approaches, *dynamic* methods aim to incorporate data mixing within a more adaptive optimization-based framework. PuzzleMix variants (Kim et al., 2020; 2021) introduce combinatorial optimization-based mixup policies in accordance with saliency maximization. SuperMix variants (Dabouei et al., 2021; Walawalkar et al., 2020) utilize pre-trained teacher models to find smooth and optimized mixed samples. AutoMix variants (Liu et al., 2022d; Li et al., 2021) first reformulate the mixup framework in an online automatic fashion that learns to adaptively generate mixing policies in an end-to-end manner.

### 2.3 LABEL MIXING

Apart from sample mixing, Mixup (Zhang et al., 2018) and CutMix (Yun et al., 2019) are two widely recognized label mixing techniques, both of which are *static*. Recently, there has been a notable emphasis among researchers on enhancing label mixup policies, attaining more favorable performance upon certain sample mixing policies. Based on Transformers, TransMix variants (Chen et al., 2022; Liu et al., 2022a; Choi et al., 2022; Chen et al., 2023) utilize the class token and attention maps to adjust the mixing ratio. A decouple learning objective (Liu et al., 2022c) is designed to help models focus on hard mixing samples, which can be plugged into popular sample mixing policies.

### 3 OPENMIXUP

This section presents an introduction to our OpenMixup codebase framework and benchmark from four key aspects: supported methods and tasks, evaluation metrics, and experimental pipeline. OpenMixup provides a unified mixup-based model design and training framework implemented in PyTorch (Paszke et al., 2019). We start with an overview of its composition. The framework design makes reference to MMClassification (Contributors, 2020a) following the OpenMMLab coding style. As shown in Figure 3, the whole training process here is fragmented into multiple components, including model architecture (`.openmixup.models`), data pre-processing (`.openmixup.datasets`), mixup policies (`.openmixup.models.utils.augments`), script tools (`.tools`) *etc.* For instance, vision models are summarized into several building blocks (*e.g.*, backbone, neck, head *etc.*) in `.openmixup.models`. This modular architecture allows researchers to easily craft models by incorporating different components as needed. With the help of configuration files in `.configs`, users can tailor specialized visual classification models and their associated training strategies with ease. In addition, benchmark configuration (`.benchmarks`) and benchmarking results (`.tools.model_zoos`) are also provided in OpenMixup. Benchmarking details are discussed below.

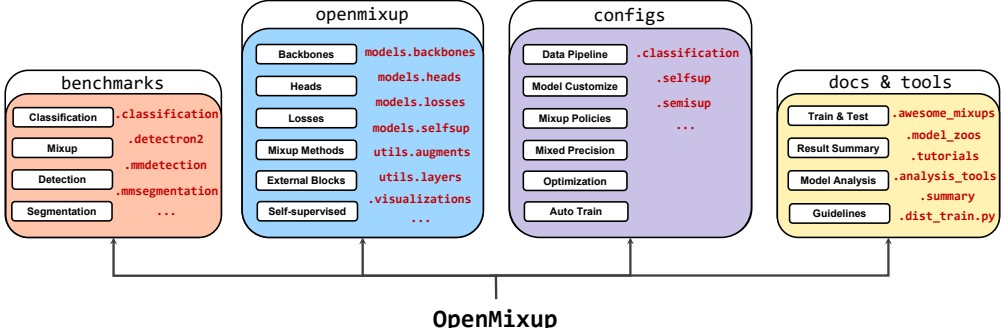

Figure 3: Overview of codebase framework of OpenMixup benchmark. (1) `benchmarks` provide benchmark results and corresponding config files for mixup classification and transfer learning. (2) `openmixup` contains source codes of supported methods. (3) `configs` is responsible for customizing setups of the methods, networks, datasets, and training pipelines. (4) `docs & tools` contains paper lists of popular methods, user documentation, and practical tools.

## 3.1 SUPPORTED METHODS

For supervised visual classification, OpenMixup has implemented 16 representative mixup augmentation algorithms and 19 convolutional neural network and Transformer model architectures (gathered in `.openmixup.models`) on 12 diverse image datasets. We summarize these methods in Table 1, where we also provide their corresponding conference/journal and the types of their sample and label mixing policies. For sample mixing, Mixup (Zhang et al., 2018) and ManifoldMix (Verma et al., 2019)perform *hand-crafted* convex interpolation. CutMix (Yun et al., 2019), SmoothMix (ha Lee et al., 2020), GridMix (Baek et al., 2021) and ResizeMix (Qin et al., 2023) implement hand-crafted cutting policy. Fmix (Harris et al., 2020) uses Fourier-guided cutting. AttentiveMix (Walawalkar et al., 2020) and SaliencyMix (Uddin et al., 2020) apply pretraining-guided and saliency-guided cutting, respectively. As for the *dynamic* methods, PuzzleMix (Kim et al., 2020) and AlignMix (Venkataramanan et al., 2022) utilize optimal-transported cutting and interpolation methods. AutoMix (Liu et al., 2022d) and SAMix (Li et al., 2021) perform online end-to-end learnable cutting-based approaches. As for the label mixing, most methods apply Mixup (Zhang et al., 2018) or CutMix (Yun et al., 2019), while the latest mixup methods for visual transformers (TransMix (Chen et al., 2022), TokenMix (Liu et al., 2022a), and SMMix (Chen et al., 2023)), as well as DecoupledMix (Liu et al., 2022c) exploit attention maps and a decoupled framework respectfully instead, which incorporate CutMix variants as its sample mixing strategy.

## 3.2 SUPPORTED TASKS

We provide detailed descriptions of the 12 open-source datasets as shown in Table 2. These datasets can be classified into four categories below: **(1) Small-scale classification**: We conduct benchmarking studies on small-scale datasets to provide an accessible benchmarking reference. CIFAR-10/100 (Krizhevsky et al., 2009) consists of 60,000 color images in $32\times32$ resolutions. Tiny-ImageNet (Tiny) (Chrabaszcz et al., 2017) and STL-10 (Coates et al., 2011) are two re-scale versions of ImageNet-1K in the size of $64\times64$ and $96\times96$. FashionMNIST (Xiao et al., 2017) is the advanced version of MNIST, which contains gray-scale images of clothing. **(2) Large-scale classification**: The large-scale dataset is employed to evaluate mixup algorithms against the most standardized procedure, which can also support the prevailing ViT architecture. ImageNet-1K (IN-1K) (Russakovsky et al., 2015) is a well-known challenging dataset for image classification with 1000 classes. **(3) Fine-grained classification**: To investigate the effectiveness of mixup methods in complex inter-

Table 2: The detailed information of supported image classification datasets in OpenMixup.

| Datasets | Category | Source | Classes | Resolution | Train images | Test images |
|---|---|---|---|---|---|---|
| CIFAR-10 (Krizhevsky et al., 2009) | Iconic | link | 10 | $32\times32$ | 50,000 | 10,000 |
| CIFAR-100 (Krizhevsky et al., 2009) | Iconic | link | 100 | $32\times32$ | 50,000 | 10,000 |
| FashionMNIST (Xiao et al., 2017) | Gray-scale | link | 10 | $28\times28$ | 50,000 | 10,000 |
| STL-10 (Coates et al., 2011) | Iconic | link | 10 | $96\times96$ | 50,00 | 8,000 |
| Tiny-ImageNet (Chrabaszcz et al., 2017) | Iconic | link | 200 | $64\times64$ | 10,000 | 10,000 |
| ImageNet-1K (Russakovsky et al., 2015) | Iconic | link | 1000 | $469\times387$ | 1,281,167 | 50,000 |
| CUB-200-2011 (Wah et al., 2011) | Fine-grained | link | 200 | $224\times224$ | 5,994 | 5,794 |
| FGVC-Aircraft (Maji et al., 2013) | Fine-grained | link | 100 | $224\times224$ | 6,667 | 3,333 |
| iNaturalist2017 Horn et al. (2018) | Fine-grained & longtail | link | 5089 | $224\times224$ | 579,184 | 95,986 |
| iNaturalist2018 Horn et al. (2018) | Fine-grained & longtail | link | 8142 | $224\times224$ | 437,512 | 24,426 |
| Places205 (Zhou et al., 2014) | Scenic | link | 205 | $224\times224$ | 2,448,873 | 41,000 |

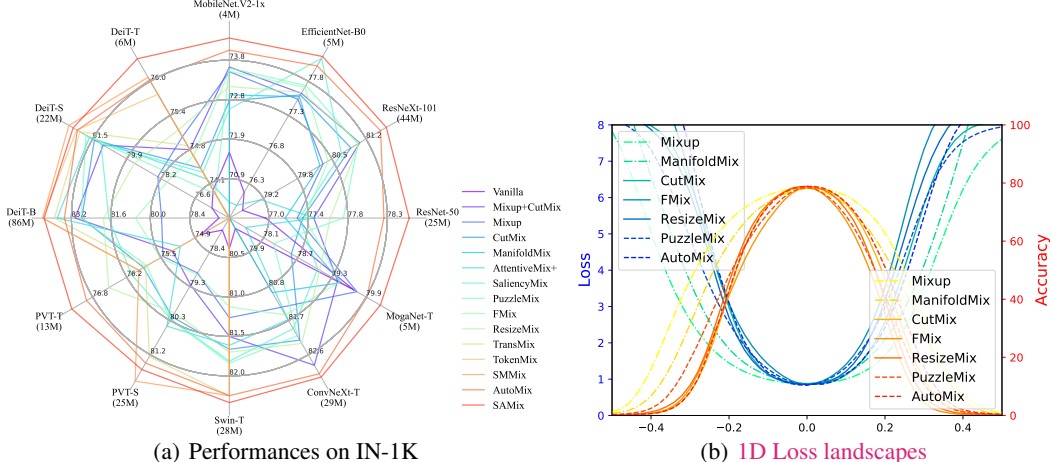

(a) Performances on IN-1K

(b) 1D Loss landscapes

Figure 4: Visualization of various backbones performances and loss landscapes (1D) for multiple mixup methods on ImageNet-1K. (a) Radar plots of top-1 accuracy using various network architectures. (b) The training loss and validation top-1 accuracy are plotted with ResNet-50, showing that the dynamic Mixup methods achieve deeper and wider loss landscapes than the static variants.

class relationships and long-tail scenarios, we conduct a comprehensive evaluation of fine-grained classification datasets, which can also be classified into small-scale and large-scale scenarios. (i) *Small-scale scenarios*: The datasets for small-scale fine-grained evaluation scenario are CUB-200-2011 (CUB) (Wah et al., 2011) and FGVC-Aircraft (Aircraft) (Maji et al., 2013), which contains a total of 200 wild bird species and 100 classes of airplanes. (ii) *Large-scale scenarios*: The datasets for large-scale fine-grained evaluation scenarios are iNaturalist2017 (iNat2017) (Horn et al., 2018) and iNaturalist2018 (iNat2018) (Horn et al., 2018), which contain 5,089 and 8,142 natural categories. Both the iNat2017 and iNat2018 own 7 major categories and are also long-tail datasets with scenic images (*i.e.*, the fore-ground target is in large backgrounds). **(4) Scenic classification**: We perform the large-scale scenic classification evaluation to investigate the consistent superiority of mixup augmentations in complex non-iconic scenarios on Places205 (Zhou et al., 2014).

## 3.3 EVALUATION METRICS

We comprehensively evaluate the beneficial properties of mixup augmentation algorithms on the aforementioned tasks through the use of various metrics and visualization analysis tools in a rigorous manner. The evaluation methodologies can be classified into two distinct divisions, namely performance metric and empirical analysis. As for the performance metrics, classification accuracy and robustness against corruption are two performance indicators examined. For empirical analysis, experiments on calibrations, CAM visualization, loss landscape, the plotting of training loss, and validation accuracy curves are conducted. The utilization of these approaches is contingent upon their distinct properties, enabling their deployment for designated purposes.

**Performance Metric.** **(1) Accuracy and training costs**: We adopt top-1 accuracy, total training hours, and GPU memory to evaluate the classification performance and training overheads of certain methods. **(2) Robustness**: We evaluate the robustness against corruptions of mixup variants on CIFAR-100-C and ImageNet-C (Russakovsky et al., 2015), which is designed for evaluating the corruption robustness and provides 19 different corruptions, *e.g.*, noise and blur *etc.* **(3) Transfer to object detection**: We also evaluate transferable abilities on object detection tasks based on Faster R-CNN (Ren et al., 2015) and Mask R-CNN (He et al., 2017) on COCO *train2017* (Lin et al., 2014), initializing with trained models on ImageNet-1K.

**Empirical Analysis.** **(1) Calibrations**: To verify the calibration ability to exist mixup methods, we evaluate them by the expected calibration error (ECE) on CIFAR-100 (Krizhevsky et al., 2009), *i.e.*, the absolute discrepancy between accuracy and confidence. **(2) CAM visualization**: We utilize mixed sample visualization, a series of CAM variants (Chattopadhyay et al., 2018; Muhammad & Yeasin, 2020) (*e.g.*, Grad-CAM (Selvaraju et al., 2019)) to directly analyze the classification accuracy and especially the localization capabilities of mixup augmentation algorithms through top-1 top-2 accuracy predicted targets. **(3) Loss landscape**: We apply loss landscape evaluation (Li et al., 2018) to further analyze the degree of loss smoothness. **(4) Training loss and accuracy curve**: We utilize the plot of training loss and validation accuracy curves to analyze the training stability, the property of preventing over-fitting, and the convergence speed of mixup augmentations.

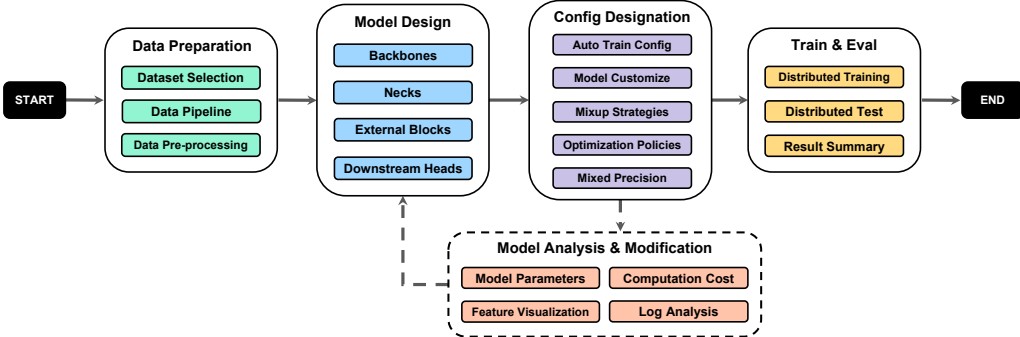

Figure 5: Overview of the experimental pipeline in OpenMixup codebase.

## 3.4 EXPERIMENTAL PIPELINE OF OPENMIXUP CODEBASE

Figure 5 offers a unified training pipeline in our OpenMixup, sharing a comparable workflow for different classification tasks. We take classification tasks as an instance to illustrate the training procedure. Firstly, users should go through the supported data pipeline and select the dataset and pre-processing techniques. Secondly, `openmixup.models` serves as a model architecture component for building desired methods. Thirdly, it is undemanding to designate the supported datasets, mixup augmentation strategies, model architectures, and optimization schedules under `.configs.classification` with Python configuration files to customize a desired setting. Afterward, `.tools` provides hardware support distributed training to execute the confirmed training process in configs. Apart from that, there are also various utility functionalities given in `.tools` (*e.g.*, feature visualization, model analysis, result summarization). Refer to online user documents at `https://openmixup.readthedocs.io/en/latest/` for detailed guidelines (*e.g.*, installation and getting started instructions), benchmark results, awesome lists of related work, *etc*.

## 4 EXPERIMENT AND ANALYSIS

We conduct essential benchmarking experiments of image classification on various scenarios with diverse evaluation metrics. For a fair comparison, grid search is performed for the shared hyper-parameter $\alpha \in \{0.1, 0.2, 0.5, 1, 2, 4\}$ of supported mixup variants while the rest of the hyper-parameters follow the original papers. Vanilla denotes the classification baseline without any mixup augmentations. All experiments are conducted on Ubuntu workstations with Tesla V100 or NVIDIA A100 GPUs and report the *mean* results of three trials. Appendix B provides more classification results, and Appendix B.4 contains transfer learning results.

**Small-scale Benchmarks.** We first provide standard mixup image classification benchmarks on five small datasets with two settings. (a) The classical settings with the CIFAR version of ResNet variants (He et al., 2016; Xie et al., 2017), *i.e.*, replacing the $7 \times 7$ convolution and MaxPooling by a $3 \times 3$ convolution. We use $32 \times 32$, $64 \times 64$, and $28 \times 28$ input resolutions for CIFAR-10/100, Tiny-ImageNet, and FashionMNIST, while using the normal ResNet for STL-10. We train models for multiple epochs from the stretch with SGD optimizer and a batch size of 100, as shown in Table 3 and Appendix B.2. (b) The modern settings following DeiT (Touvron et al., 2021) on CIFAR-100, using $224 \times 224$ and $32 \times 32$ resolutions for Transformers (DeiT-S (Touvron et al., 2021) and Swin-T (Liu et al., 2021)) and ConvNeXt-T (Liu et al., 2022b) in Table A7.

**Standard ImageNet-1K Benchmarks.** For visual augmentation and network architecture communities, ImageNet-1K is a well-known standard dataset. We support three popular training recipes: (a) PyTorch-style (He et al., 2016) setting for classical CNNs; (b) timm RSB A2/A3 (Wightman et al., 2021) settings; (c) DeiT (Touvron et al., 2021) setting for ViT-based models. Evaluation is performed on 224×224 resolutions with `CenterCrop`. Popular network architectures are considered: ResNet (He et al., 2016), Wide-ResNet (Zagoruyko & Komodakis, 2016), ResNeXt (Xie et al., 2017), MobileNet.V2 (Sandler et al., 2018), EfficientNet (Tan & Le, 2019), DeiT (Touvron et al., 2021), Swin (Liu et al., 2021), ConvNeXt (Liu et al., 2022b), and MogaNet (Li et al., 2022). Refer to Appendix A for implementation details. In Table A2 and Table 4, we report the *mean* performance of three trials where the *median* of top-1 test accuracy in the last 10 epochs is recorded for each trial.

**Benchmarks on Fine-grained and Scenis Scenarios.** We further provide benchmarks on three downstream classification scenarios in 224×224 resolutions with ResNet architectures: (a) Transfer

Table 3: Top-1 accuracy (%) on CIFAR-10/100 and Tiny based on ResNet (R), Wide-ResNet (WRN), ResNeXt (RX).

| Datasets | CIFAR-10 | CIFAR-100 | Tiny |
|---|---|---|---|
| Backbones | R-18 | WRN-28-8 | RX-50 |
| Epochs | 800 ep | 800 ep | 400 ep |
| Vanilla | 95.50 | 81.63 | 65.04 |
| Mixup | 96.62 | 82.82 | 66.36 |
| CutMix | 96.68 | 84.45 | 66.47 |
| ManifoldMix | 96.71 | 83.24 | 67.30 |
| SmoothMix | 96.17 | 82.09 | 68.61 |
| AttentiveMix | 96.63 | 84.34 | 67.42 |
| SaliencyMix | 96.20 | 84.35 | 66.55 |
| FMix | 96.18 | 84.21 | 65.08 |
| GridMix | 96.56 | 84.24 | 69.12 |
| ResizeMix | 96.76 | 84.87 | 65.87 |
| PuzzleMix | 97.10 | 85.02 | 67.83 |
| Co-Mixup | 97.15 | 85.05 | 68.02 |
| AlignMix | 97.05 | 84.87 | 68.74 |
| AutoMix | 97.34 | 85.18 | 70.72 |
| SAMix | **97.50** | **85.50** | **72.18** |
| Decoupled | 96.95 | 84.88 | 67.46 |

Table 4: Top-1 accuracy (%) on ImageNet-1K using PyTorch-style, RSB A2/A3, and DeiT settings based on CNN and Transformer architectures, including ResNet (R), MobileNet.V2 (Mob.V2), DeiT-S, and Swin-T.

| Backbones | R-50 | R-50 | Mob.V2 1x | DeiT-S | Swin-T |
|---|---|---|---|---|---|
| Epochs | 100 ep | 100 ep | 300 ep | 300 ep | 300 ep |
| Settings | PyTorch | RSB A3 | RSB A2 | DeiT | DeiT |
| Vanilla | 76.83 | 77.27 | 71.05 | 75.66 | 80.21 |
| Mixup | 77.12 | 77.66 | 72.78 | 77.72 | 81.01 |
| CutMix | 77.17 | 77.62 | 72.23 | 80.13 | 81.23 |
| DeiT / RSB | 77.35 | 78.08 | 72.87 | 79.80 | 81.20 |
| ManifoldMix | 77.01 | 77.78 | 72.34 | 78.03 | 81.15 |
| AttentiveMix | 77.28 | 77.46 | 70.30 | 80.32 | 81.29 |
| SaliencyMix | 77.14 | 77.93 | 72.07 | 79.88 | 81.37 |
| FMix | 77.19 | 77.76 | 72.79 | 80.45 | 81.47 |
| ResizeMix | 77.42 | 77.85 | 72.50 | 78.61 | 81.36 |
| PuzzleMix | 77.54 | 78.02 | 72.85 | 77.37 | 79.60 |
| AutoMix | 77.91 | 78.44 | 73.19 | 80.78 | 81.80 |
| SAMix | **78.06** | **78.64** | **73.42** | 80.94 | **81.87** |
| TransMix | - | - | - | 80.68 | 81.80 |
| SMMix | - | - | - | **81.10** | 81.80 |

(a) DeiT-S on IN-1K     (b) DeiT-S on CIFAR-100     (c) ConvNeXt-T on CIFAR-100

Figure 6: Performance, total training time (hours), and GPU memory (G) trade-off paradigms. (a) is based on DeiT-S on ImageNet-1K and (b)(c) are based on DeiT-S and ConvNeXt-T on CIFAR-100.

learning on CUB-200 and FGVC-Aircraft. (b) Fine-grained classification on iNat2017 and iNat2018. (c) Scenic classification on Places205, as shown in Appendix B.3 and Table A10.

**Empirical Observations and Insights.** Empirical analysis of the benchmarking results is presented to gain insightful observations and an in-depth understanding of mixup visual classification.

(A) **Which mixup method should I choose?** After a large number of experiments and empirical analysis, we integrate the results from multiple perspectives (please refer to appendix) and eventually provide a comprehensive ranking based on *performance*, *applicability*, and the *overall* capacity. As shown in Table 5, from the *performance* angle, the *online-optimizable* SAMix and AutoMix stand out as the top two. SMMix and TransMix are close behind. However, in terms of *applicability*, *handcrafted* mixup variants are far ahead of the learning-based ones. Collectively, DeiT (Mixup+CutMix), SAMix, and SMMix emerge as the three most preferable mixup methods.

(B) **Generalizability over datasets:** We present a clear and comprehensive mixup performance radar chart in Figure 1. Combining the trade-off results in Fiugre 6, we can conclude that *dynamic* mixup consistently yields better performance compared to the others, showing its impressive generalizability over different visual classification datasets. However, these *dynamic* approaches necessitate meticulous tuning, which incurs considerable training costs. By contrast, *static* mixup exhibits significant performance variance across different scenarios, which demonstrates its poor generalizability. For instance, Mixup (Zhang et al., 2018) and CutMix (Yun et al., 2019) perform even worse than the baseline without mixup on Place205 and FGVC-Aircraft, respectively.

(C) **Generalizability over backbones:** As shown in Figure 4 and Figure 6, we provide extensive evaluations on ImageNet-1K based on different types of backbones and mixup methods. As a result, *Dynamic* mixup achieves better performance than *static* mixup in general while showing more favorable robustness against backbone selection. Notably, the *online-optimizable* SAMix and AutoMix show impressive robustness against backbone selection, which may reveal the hierarchical superiority of the online training paradigm in a sense. However, mixup methods that

Table 5: Rankings of various mixup augmentations as take-home conclusions for practical usages.

| | Mixup | CutMix | DeiT Smooth | GridMix | ResizeMix | Manifold | FMix | Attentive | Saliency | PuzzleMix | AlignMix | AutoMix | SAMix | TransMix | SMMix |
|---|---|---|---|---|---|---|---|---|---|---|---|---|---|---|---|
| Performance | 13 | 11 | 5 | 16 | 15 | 8 | 12 | 14 | 7 | 9 | 6 | 10 | 2 | 1 | 4 | 3 |
| Applicability | 1 | 1 | 1 | 1 | 1 | 1 | 1 | 1 | 3 | 1 | 4 | 2 | 7 | 6 | 5 | 5 |
| Overall | 8 | 6 | 1 | 11 | 10 | 4 | 7 | 9 | 5 | 5 | 5 | 6 | 4 | 2 | 4 | 3 |

Figure 7: Visualization of class activation mapping (CAM) (Selvaraju et al., 2019) for top-1 and top-2 predicted classes of supported mixup methods with ResNet-50 on ImageNet-1K. Comparing the first and second rows, we observe that saliency-guided or dynamic mixup approaches (*e.g.*, PuzzleMix and SAMix) localize the target regions better than the static methods (*e.g.*, Mixup and ResizeMix).

solely designed for ViTs (*e.g.*, TransMix (Chen et al., 2022) and TokenMix (Liu et al., 2022a)) yield exceptional performance with DeiT-S and PVT-S backbone yet exhibit intense sensitivity to model scale (*e.g.*, with PVT-T), which may limit their application potentials for specific issues.

(D) **Compatibility:** As illustrated in Figure 4, *Dynamic* mixup shows better compatibility than *static* mixup. Nevertheless, recent ViT-only *dynamic* mixup methods exhibit poor model compatibility. Surprisingly, *static* Mixup (Zhang et al., 2018) exhibits favorable compatibility with newly-emerged efficient networks like MogaNet (Li et al., 2022). Notably, *Static* CutMix (Yun et al., 2019) fits well with both modern CNNs (*e.g.*, ConvNeXt and ResNeXt) and DeiT.

(E) **Convergence & Training Stability:** As shown in Figure 4, wider bump curves indicate greater training stability, while higher warm color bump tips are associated with better convergence. It is obvious that *dynamic* mixup owns better training stability and convergence than *static* mixup. The only exception is Mixup (Zhang et al., 2018), which exhibits better training stability than most of *dynamic* mixing methods. We assume this arises from its convex interpolation that principally prioritizes training stability but may lead to suboptimal outcomes.

(F) **Localizability & Downstream Transferability:** To ensure the transferability of mixup methods, we hypothesize that it is not enough to merely refer to the aforementioned aspects of the reference. It is commonly conjectured that the mixup methods with better localizability can be better transferred to detection-related downstream tasks. Thus, we visualize the class activation mapping (CAM) for top-1 and top-2 predicted classes of the supported mixup methods with ResNet-50 backbone on ImageNet-1K. As illustrated in Figure 7, SAMix and AutoMix's exceptional localizability, combined with their stellar accuracy performance and robustness, may indicate a hierarchical superiority of their online training framework on detection tasks.

## 5 CONCLUSION AND DISCUSSION

**Contributions:** In this work, we presents the *first* comprehensive mixup visual classification benchmark, called OpenMixup, which also offers a unified modular mixup-based model design and training codebase framework for visual representation learning. Extensive experiments on both *static* and *dynamic* mixup baselines are conducted across various visual classification datasets. Empirical analysis provides an in-depth investigation, offering valuable observations, insights and take-home messages that contribute to a systematic comprehension of mixups. We anticipate that this study can not only provide a standardized mixup benchmark and a practical codebase platform but contribute to both industrial usage and future research progress in the mixup community. We also encourage practitioners and researchers to extend their valuable feedback to us and contribute to our OpenMixup codebase for building a more constructive mixup representation learning framework through GitHub.

**Limitations and Future Works:** The scope of this work is largely limited to the most representative visual classification tasks, albeit we provide Transfer Learning to Object Detection and Semantic Segmentation tasks. More detailed benchmarking results are provided in the supplementary materials for a better evaluation of mixup methods. The OpenMixup codebase framework can be extended to other tasks and scenarios, such as object detection, semantic segmentation, and self-supervised visual representation learning. We believe this work as the *first* mixup benchmarking study can give the research community a kick-start, and we plan to extend our work in these directions in the future.

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

## SUPPLEMENT MATERIAL

In supplement material, we provide implementation details and more benchmark results of image classification with mixup augmentations implemented in `OpenMixup` on various datasets.

## A    IMPLEMENTATION DETAILS

### A.1    SETUP OPENMIXUP

As provided in the supplementary material, we simply introduce the installation and data preparation for OpenMixup, detailed in 'docs/en/latest/install.md'. Assuming the PyTorch environment has already been installed, users can easily reproduce the environment with the source code by executing the following commands:

```
conda activate openmixup
pip install openmim
mim install mmcv-full
\# put the source code here
cd openmixup
python setup.py develop  \# or "pip install -e ."
```

Executing the instructions above, OpenMixup will be installed as the development mode, *i.e.*, any modifications to the local source code take effect, and can be used as a python package. Then, users can download the datasets and the released meta files and symlink them to the dataset root (`$OpenMixup/data`).

### A.2    TRAINING SETTINGS OF IMAGE CLASSIFICATION

**Large-scale Datasets.**    Table A1 illustrates three popular training settings on large-scaling datasets like ImageNet-1K in detail: (1) PyTorch-style (Paszke et al., 2019). (2) DeiT (Touvron et al., 2021). (3) RSB A2/A3 (Wightman et al., 2021). Notice that the step learning rate decay strategy is replaced by Cosine Scheduler (Loshchilov & Hutter, 2016), and `ColorJitter` as well as `PCA lighting` are removed in PyTorch-style setting for better performances. DeiT and RSB settings adopt advanced augmentation and regularization techniques for Transformers, while RSB A3 is a simplified setting for fast training on ImageNet-1K. For a fare comparison, we search the optimal hyper-parameter $\alpha$ in $Beta(\alpha, \alpha)$ from $\{0.1, 0.2, 0.5, 1, 2, 4\}$ for compared methods while the rest of the hyper-parameters follow the original papers.

**Small-scale Datasets.**    We also provide two experimental settings on small-scale datasets: (a) Following the common setups (He et al., 2016; Yun et al., 2019) on small-scale datasets like CIFAR-10/100, we train 200/400/800/1200 epochs from stretch based on CIFAR version of ResNet variants (He et al., 2016), *i.e.*, replacing the $7 \times 7$ convolution and MaxPooling by a $3 \times 3$ convolution. As for the data augmentation, we apply `RandomFlip` and `RandomCrop` with 4 pixels padding for $32 \times 32$ resolutions. The testing image size is $32 \times 32$ (no `CenterCrop`). The basic training settings include: SGD optimizer with SGD weight decay of 0.0001, a momentum of 0.9, a batch size of 100, and a basic learning rate is 0.1 adjusted by Cosine Scheduler (Loshchilov & Hutter, 2016). (b) We also provide modern training settings following DeiT (Touvron et al., 2021), while using $224 \times 224$ and $32 \times 32$ resolutions for Transformer and CNN architectures. We only changed the batch size to 100 for CIFAR-100 and borrowed other settings the same as DeiT on ImageNet-1K.

## B    MIXUP IMAGE CLASSIFICATION BENCHMARKS

### B.1    MIXUP BENCHMARKS ON IMAGENET-1K

**PyTorch-style training settings**    The benchmark results are illustrated in Table A2. Notice that we adopt $\alpha = 0.2$ for some cutting-based mixups (CutMix, SaliencyMix, FMix, ResizeMix) based on ResNet-18 since ResNet-18 might be under-fitted on ImageNet-1k.

**DeiT training setting**    Table A3 shows the benchmark results following DeiT training setting. Experiment details refer to Sec. A.2. Notice that the performances of transformer-based architectures

Table A1: Ingredients and hyper-parameters used for ImageNet-1K training settings.

| Procedure | PyTorch | DeiT | RSB A2 | RSB A3 |
|---|---|---|---|---|
| Train Res | 224 | 224 | 224 | 160 |
| Test Res | 224 | 224 | 224 | 224 |
| Test crop ratio | 0.875 | 0.875 | 0.95 | 0.95 |
| Epochs | 100/300 | 300 | 300 | 100 |
| Batch size | 256 | 1024 | 2048 | 2048 |
| Optimizer | SGD | AdamW | LAMB | LAMB |
| LR | 0.1 | $1 \times 10^{-3}$ | $5 \times 10^{-3}$ | $8 \times 10^{-3}$ |
| LR decay | cosine | cosine | cosine | cosine |
| Weight decay | $10^{-4}$ | 0.05 | 0.02 | 0.02 |
| Warmup epochs | ✗ | 5 | 5 | 5 |
| Label smoothing $\epsilon$ | ✗ | 0.1 | ✗ | ✗ |
| Dropout | ✗ | ✗ | ✗ | ✗ |
| Stoch. Depth | ✗ | 0.1 | 0.05 | ✗ |
| Repeated Aug | ✗ | ✓ | ✓ | ✗ |
| Gradient Clip. | ✗ | 1.0 | ✗ | ✗ |
| H. flip | ✓ | ✓ | ✓ | ✓ |
| RRC | ✓ | ✓ | ✓ | ✓ |
| Rand Augment | ✗ | 9/0.5 | 7/0.5 | 6/0.5 |
| Auto Augment | ✗ | ✗ | ✗ | ✗ |
| Mixup alpha | ✗ | 0.8 | 0.1 | 0.1 |
| Cutmix alpha | ✗ | 1.0 | 1.0 | 1.0 |
| Erasing prob. | ✗ | 0.25 | ✗ | ✗ |
| ColorJitter | ✗ | ✗ | ✗ | ✗ |
| EMA | ✗ | ✓ | ✗ | ✗ |
| CE loss | ✓ | ✓ | ✗ | ✗ |
| BCE loss | ✗ | ✗ | ✓ | ✓ |
| Mixed precision | ✗ | ✗ | ✓ | ✓ |

Table A2: Top-1 accuracy (%) of image classification based on ResNet variants on ImageNet-1K using PyTorch-style 100-epoch and 300-epoch training procedures.

| Methods | Beta $\alpha$ | PyTorch 100 epochs | | | | | PyTorch 300 epochs | | | |
|---|---|---|---|---|---|---|---|---|---|---|
| | | R-18 | R-34 | R-50 | R-101 | RX-101 | R-18 | R-34 | R-50 | R-101 |
| Vanilla | - | 70.04 | 73.85 | 76.83 | 78.18 | 78.71 | 71.83 | 75.29 | 77.35 | 78.91 |
| MixUp | 0.2 | 69.98 | 73.97 | 77.12 | 78.97 | 79.98 | 71.72 | 75.73 | 78.44 | 80.60 |
| CutMix | 1 | 68.95 | 73.58 | 77.17 | 78.96 | 80.42 | 71.01 | 75.16 | 78.69 | 80.59 |
| ManifoldMix | 0.2 | 69.98 | 73.98 | 77.01 | 79.02 | 79.93 | 71.73 | 75.44 | 78.21 | 80.64 |
| SaliencyMix | 1 | 69.16 | 73.56 | 77.14 | 79.32 | 80.27 | 70.21 | 75.01 | 78.46 | 80.45 |
| FMix | 1 | 69.96 | 74.08 | 77.19 | 79.09 | 80.06 | 70.30 | 75.12 | 78.51 | 80.20 |
| ResizeMix | 1 | 69.50 | 73.88 | 77.42 | 79.27 | 80.55 | 71.32 | 75.64 | 78.91 | 80.52 |
| PuzzleMix | 1 | 70.12 | 74.26 | 77.54 | 79.43 | 80.53 | 71.64 | 75.84 | 78.86 | 80.67 |
| AutoMix | 2 | 70.50 | 74.52 | 77.91 | 79.87 | 80.89 | 72.05 | 76.10 | 79.25 | 80.98 |
| SAMix | 2 | **70.83** | **74.95** | **78.06** | **80.05** | **80.98** | **72.27** | **76.28** | **79.39** | **81.10** |

are more difficult to reproduce than ResNet variants, and the mean of the best performance in 3 trials is reported as their original paper.

**RSB A2/A3 training settings** The RSB A2/A3 benchmark results based on ResNet-50, EfficientNet-B0, and MobileNet.V2 are illustrated in Table A4. Training 300/100 epochs with the BCE loss on ImageNet-1k, RSB A3 is a fast training setting, while RSB A2 can exploit the full representation ability of ConvNets. Notice that the RSB settings employ Mixup with $\alpha = 0.1$ and CutMix with $\alpha = 1.0$. We report the mean of top-1 accuracy in the last 5/10 training epochs for 100/300 epochs.

## B.2 SMALL-SCALE CLASSIFICATION BENCHMARKS

To facilitate fast research on mixup augmentations, we benchmark mixup image classification on CIFAR-10/100 and Tiny-ImageNet with two settings.

Table A3: Top-1 accuracy (%) on ImageNet-1K based on popular Transformer-based architectures using DeiT-S training settings. Notice that † denotes reproducing results with the official implementation, while other results are implemented with OpenMixup. TransMix, TokenMix, and SMMix are specially designed for Transformers.

| Methods | $\alpha$ | DeiT-T | DeiT-S | DeiT-B | PVT-T | PVT-S | Swin-T | ConvNeXt-T | MogaNet-T |
|---|---|---|---|---|---|---|---|---|---|
| Vanilla | - | 73.91 | 75.66 | 77.09 | 74.67 | 77.76 | 80.21 | 79.22 | 79.25 |
| DeiT | 0.8, 1 | 74.50 | 79.80 | 81.83 | 75.10 | 78.95 | 81.20 | 82.10 | 79.02 |
| MixUp | 0.2 | 74.69 | 77.72 | 78.98 | 75.24 | 78.69 | 81.01 | 80.88 | 79.29 |
| CutMix | 0.2 | 74.23 | 80.13 | 81.61 | 75.53 | 79.64 | 81.23 | 81.57 | 78.37 |
| ManifoldMix | 0.2 | - | - | - | - | - | - | 80.57 | 79.07 |
| AttentiveMix+ | 2 | 74.07 | 80.32 | 82.42 | 74.98 | 79.84 | 81.29 | 81.14 | 77.53 |
| SaliencyMix | 0.2 | 74.17 | 79.88 | 80.72 | 75.71 | 79.69 | 81.37 | 81.33 | 78.74 |
| FMix | 0.2 | 74.41 | 77.37 | | 75.28 | 78.72 | 79.60 | 81.04 | 79.05 |
| ResizeMix | 1 | 74.79 | 78.61 | 80.89 | 76.05 | 79.55 | 81.36 | 81.64 | 78.77 |
| PuzzleMix | 1 | 73.85 | 80.45 | 81.63 | 75.48 | 79.70 | 81.47 | 81.48 | 78.12 |
| AutoMix | 2 | 75.52 | 80.78 | 82.18 | 76.38 | 80.64 | 81.80 | 82.28 | 79.43 |
| SAMix | 2 | **75.83** | **80.94** | 82.85 | **76.60** | 80.78 | **81.87** | **82.35** | **79.62** |
| TransMix | 0.8, 1 | 74.56 | 80.68 | 82.51 | 75.50 | 80.50 | 81.80 | - | - |
| TokenMix† | 0.8, 1 | 75.31 | 80.80 | **82.90** | 75.60 | - | 81.60 | - | - |
| SMMix | 0.8, 1 | 75.56 | 81.10 | 82.90 | 75.60 | **81.03** | 81.80 | - | - |

Table A4: Top-1 accuracy (%) on ImageNet-1K based on classical ConvNets using RSB A2/A3 training settings, including ResNet, EfficientNet, and MobileNet.V2.

| Backbones Settings | $Beta$ $\alpha$ | R-50 A3 | R-50 A2 | Eff-B0 A3 | Eff-B0 A2 | Mob.V2 1× A3 | Mob.V2 1× A2 |
|---|---|---|---|---|---|---|---|
| RSB | 0.1, 1 | 78.08 | 79.80 | 74.02 | 77.26 | 69.86 | 72.87 |
| MixUp | 0.2 | 77.66 | 79.39 | 73.87 | 77.19 | 70.17 | 72.78 |
| CutMix | 0.2 | 77.62 | 79.38 | 73.46 | 77.24 | 69.62 | 72.23 |
| ManifoldMix | 0.2 | 77.78 | 79.47 | 73.83 | 77.22 | 70.05 | 72.34 |
| AttentiveMix+ | 2 | 77.46 | 79.34 | 72.16 | 75.95 | 67.32 | 70.30 |
| SaliencyMix | 0.2 | 77.93 | 79.42 | 73.42 | 77.67 | 69.69 | 72.07 |
| FMix | 0.2 | 77.76 | 79.05 | 73.71 | 77.33 | 70.10 | 72.79 |
| ResizeMix | 1 | 77.85 | 79.94 | 73.67 | 77.27 | 69.94 | 72.50 |
| PuzzleMix | 1 | 78.02 | 79.78 | 74.10 | 77.35 | 70.04 | 72.85 |
| AutoMix | 2 | 78.44 | 80.28 | 74.61 | 77.58 | 71.16 | 73.19 |
| SAMix | 2 | **78.64** | **80.40** | **75.28** | **77.69** | **71.24** | **73.42** |

Table A5: Top-1 accuracy (%) on CIFAR-10 training 200, 400, 800, 1200 epochs based on ResNet (R), Wide-ResNet (WRN), ResNeXt-32x4d (RX).

| Backbones Epochs | Beta $\alpha$ | R-18 200 ep | R-18 400 ep | R-18 800 ep | R-18 1200ep | Beta $\alpha$ | RX-50 200 ep | RX-50 400 ep | RX-50 800 ep | RX-50 1200ep |
|---|---|---|---|---|---|---|---|---|---|---|
| Vanilla | - | 94.87 | 95.10 | 95.50 | 95.59 | - | 95.92 | 95.81 | 96.23 | 96.26 |
| MixUp | 1 | 95.70 | 96.55 | 96.62 | 96.84 | 1 | 96.88 | 97.19 | 97.30 | 97.33 |
| CutMix | 0.2 | 96.11 | 96.13 | 96.68 | 96.56 | 0.2 | 96.78 | 96.54 | 96.60 | 96.35 |
| ManifoldMix | 2 | 96.04 | 96.57 | 96.71 | 97.02 | 2 | 96.97 | 97.39 | 97.33 | 97.36 |
| SmoothMix | 0.5 | 95.29 | 95.88 | 96.17 | 96.17 | 0.2 | 95.87 | 96.37 | 96.49 | 96.77 |
| AttentiveMix+ | 2 | 96.21 | 96.45 | 96.63 | 96.49 | 2 | 96.84 | 96.91 | 96.87 | 96.62 |
| SaliencyMix | 0.2 | 96.05 | 96.42 | 96.20 | 96.18 | 0.2 | 96.65 | 96.89 | 96.70 | 96.60 |
| FMix | 0.2 | 96.17 | 96.53 | 96.18 | 96.01 | 0.2 | 96.72 | 96.76 | 96.76 | 96.10 |
| GridMix | 0.2 | 95.89 | 96.33 | 96.56 | 96.58 | 0.2 | 97.18 | 97.30 | 96.40 | 95.79 |
| ResizeMix | 1 | 96.16 | 96.91 | 96.76 | 97.04 | 1 | 97.02 | 97.38 | 97.21 | 97.36 |
| PuzzleMix | 1 | 96.42 | 96.87 | 97.10 | 97.13 | 1 | 97.05 | 97.24 | 97.37 | 97.34 |
| AutoMix | 2 | 96.59 | 97.08 | 97.34 | 97.30 | 2 | 97.19 | 97.42 | 97.65 | 97.51 |
| SAMix | 2 | **96.67** | **97.16** | **97.50** | **97.41** | 2 | **97.23** | **97.51** | **97.93** | **97.74** |

**CIFAR-10** As elucidated in Sec. A.2, CIFAR-10 benchmarks based on CIFAR version ResNet variants follow CutMix settings, training 200/400/800/1200 epochs from stretch. As shown in Table A5, we report the median of top-1 accuracy in the last 10 training epochs.

Table A6: Top-1 accuracy (%) on CIFAR-100 training 200, 400, 800, 1200 epochs based on ResNet (R), Wide-ResNet (WRN), ResNeXt-32x4d (RX). Notice that † denotes reproducing results with the official implementation, while other results are implemented with OpenMixup.

| Backbones | Beta | R-18 | R-18 | R-18 | R-18 | RX-50 | RX-50 | RX-50 | RX-50 | WRN-28-8 |
| Epochs | $\alpha$ | 200 ep | 400 ep | 800 ep | 1200ep | 200 ep | 400 ep | 800 ep | 1200ep | 400ep |
|---|---|---|---|---|---|---|---|---|---|---|
| Vanilla | - | 76.42 | 77.73 | 78.04 | 78.55 | 79.37 | 80.24 | 81.09 | 81.32 | 81.63 |
| MixUp | 1 | 78.52 | 79.34 | 79.12 | 79.24 | 81.18 | 82.54 | 82.10 | 81.77 | 82.82 |
| CutMix | 0.2 | 79.45 | 79.58 | 78.17 | 78.29 | 81.52 | 78.52 | 78.32 | 77.17 | 84.45 |
| ManifoldMix | 2 | 79.18 | 80.18 | 80.35 | 80.21 | 81.59 | 82.56 | 82.88 | 83.28 | 83.24 |
| SmoothMix | 0.2 | 77.90 | 78.77 | 78.69 | 78.38 | 80.68 | 79.56 | 78.95 | 77.88 | 82.09 |
| SaliencyMix | 0.2 | 79.75 | 79.64 | 79.12 | 77.66 | 80.72 | 78.63 | 78.77 | 77.51 | 84.35 |
| AttentiveMix+ | 2 | 79.62 | 80.14 | 78.91 | 78.41 | 81.69 | 81.53 | 80.54 | 79.60 | 84.34 |
| FMix | 0.2 | 78.91 | 79.91 | 79.69 | 79.50 | 79.87 | 78.99 | 79.02 | 78.24 | 84.21 |
| GridMix | 0.2 | 78.23 | 78.60 | 78.72 | 77.58 | 81.11 | 79.80 | 78.90 | 76.11 | 84.24 |
| ResizeMix | 1 | 79.56 | 79.19 | 80.01 | 79.23 | 79.56 | 79.78 | 80.35 | 79.73 | 84.87 |
| PuzzleMix | 1 | 79.96 | 80.82 | 81.13 | 81.10 | 81.69 | 82.84 | 82.85 | 82.93 | 85.02 |
| Co-Mixup† | 2 | 80.01 | 80.87 | 81.17 | 81.18 | 81.73 | 82.88 | 82.91 | 82.97 | 85.05 |
| AutoMix | 2 | 80.12 | 81.78 | 82.04 | 81.95 | 82.84 | 83.32 | 83.64 | 83.80 | 85.18 |
| SAMix | 2 | **81.21** | **81.97** | **82.30** | **82.41** | **83.81** | **84.27** | **84.42** | **84.31** | **85.50** |

Table A7: Top-1 accuracy (%), GPU memory (G), and total training time (h) of 600 epochs on CIFAR-100 training 200 and 600 epochs based on DeiT-S, Swin-T, and ConvNeXt-T with the DeiT training setting. Notice that all methods are trained on a single A100 GPU to collect training times and GPU memory.

| Methods | $\alpha$ | DeiT-Small | | | | Swin-Tiny | | | | ConvNeXt-Tiny | | | |
| | | 200 ep | 600 ep | Mem. | Time | 200 ep | 600 ep | Mem. | Time | 200 ep | 600 ep | Mem. | Time |
|---|---|---|---|---|---|---|---|---|---|---|---|---|---|
| Vanilla | - | 65.81 | 68.50 | 8.1 | 27 | 78.41 | 81.29 | 11.4 | 36 | 78.70 | 80.65 | 4.2 | 10 |
| Mixup | 0.8 | 69.98 | 76.35 | 8.2 | 27 | 76.78 | 83.67 | 11.4 | 36 | 81.13 | 83.08 | 4.2 | 10 |
| CutMix | 2 | 74.12 | 79.54 | 8.2 | 27 | 80.64 | 83.38 | 11.4 | 36 | 82.46 | 83.20 | 4.2 | 10 |
| DeiT | 0.8, 1 | 75.92 | 79.38 | 8.2 | 27 | 81.25 | 84.41 | 11.4 | 36 | 83.09 | 84.12 | 4.2 | 10 |
| ManifoldMix | 2 | - | - | 8.2 | 27 | - | - | 11.4 | 36 | 82.06 | 83.94 | 4.2 | 10 |
| SmoothMix | 0.2 | 67.54 | 80.25 | 8.2 | 27 | 66.69 | 81.18 | 11.4 | 36 | 78.87 | 81.31 | 4.2 | 10 |
| SaliencyMix | 0.2 | 69.78 | 76.60 | 8.2 | 27 | 80.40 | 82.58 | 11.4 | 36 | 82.82 | 83.03 | 4.2 | 10 |
| AttentiveMix+ | 2 | 75.98 | 80.33 | 8.3 | 35 | 81.13 | 83.69 | 11.5 | 43 | 82.59 | 83.04 | 4.3 | 14 |
| FMix | 1 | 70.41 | 74.31 | 8.2 | 27 | 80.72 | 82.82 | 11.4 | 36 | 81.79 | 82.29 | 4.2 | 10 |
| GridMix | 1 | 68.86 | 74.96 | 8.2 | 27 | 78.54 | 80.79 | 11.4 | 36 | 79.53 | 79.66 | 4.2 | 10 |
| ResizeMix | 1 | 68.45 | 71.95 | 8.2 | 27 | 80.16 | 82.36 | 11.4 | 36 | 82.53 | 82.91 | 4.2 | 10 |
| PuzzleMix | 2 | 73.60 | 81.01 | 8.3 | 35 | 80.33 | 84.74 | 11.5 | 45 | 82.29 | 84.17 | 4.3 | 53 |
| AlignMix | 1 | - | - | - | - | 78.91 | 83.34 | 12.6 | 39 | 80.88 | 83.03 | 4.2 | 13 |
| AutoMix | 2 | 76.24 | 80.91 | 18.2 | 59 | 82.67 | 84.05 | 29.2 | 75 | 83.30 | 84.79 | 10.2 | 56 |
| SAMix | 2 | **77.94** | **82.49** | 21.3 | 58 | **82.70** | **84.74** | 29.3 | 75 | **83.56** | **84.98** | 10.3 | 57 |
| TransMix | 0.8, 1 | 76.17 | 79.33 | 8.4 | 28 | 81.33 | 84.45 | 11.5 | 37 | - | - | - | - |
| SMMix | 0.8, 1 | 74.49 | 80.05 | 8.4 | 28 | 81.55 | - | 11.5 | 37 | - | - | - | - |

Table A8: More evaluation metric on CIFAR-100 training 200 epochs, including top-1 accuracy (%)↑ (clean data, corruption data, and FGSM attacks) and calibration ECE (%)↓.

| Methods | $\alpha$ | DeiT-Small | | | | Swin-Tiny | | | |
| | | Clearn | Corruption | FGSM | ECE↓ | Clearn | Corruption | FGSM | ECE↓ |
|---|---|---|---|---|---|---|---|---|---|
| Vanilla | - | 65.81 | 49.31 | 20.58 | 9.48 | 78.41 | 58.20 | 12.87 | 11.67 |
| Mixup | 0.8 | 69.98 | 55.85 | 17.65 | 7.38 | 76.78 | 59.11 | 15.03 | 13.89 |
| CutMix | 2 | 74.12 | 55.08 | 12.53 | 6.18 | 80.64 | 57.73 | 18.38 | 10.95 |
| DeiT | 0.8, 1 | 75.92 | 57.36 | 18.55 | 5.38 | 81.25 | 62.21 | 15.66 | 15.68 |
| SmoothMix | 0.2 | 67.54 | 52.42 | 15.07 | 30.59 | 66.69 | 49.69 | 9.79 | 27.10 |
| SaliencyMix | 0.2 | 69.78 | 51.14 | 17.31 | 5.45 | 80.40 | 58.43 | 15.29 | 10.49 |
| AttentiveMix+ | 2 | 75.98 | 57.57 | 13.90 | 9.89 | 81.13 | 58.07 | 15.43 | 9.60 |
| FMix | 1 | 70.41 | 51.94 | 12.20 | 4.14 | 80.72 | 58.44 | 13.97 | 9.19 |
| GridMix | 1 | 68.86 | 51.11 | 8.43 | 4.09 | 78.54 | 57.78 | 11.07 | 9.37 |
| ResizeMix | 1 | 68.45 | 50.87 | 20.03 | 7.64 | 80.16 | 57.37 | 13.64 | 7.68 |
| PuzzleMix | 2 | 73.60 | 57.67 | 17.44 | 9.45 | 80.33 | 60.67 | 12.96 | 16.23 |
| AlignMix | 1 | - | - | - | - | 78.91 | 61.61 | 17.20 | **1.92** |
| AutoMix | 2 | 76.24 | 60.08 | 27.35 | 4.69 | 82.67 | **64.10** | 23.62 | 9.19 |
| SAMix | 2 | **77.94** | **61.91** | **30.35** | **4.01** | **82.70** | 62.19 | **23.66** | 7.85 |
| TransMix | 0.8, 1 | 76.17 | 59.89 | 22.48 | 8.28 | 81.33 | 62.53 | 18.90 | 16.47 |
| SMMix | 0.8, 1 | 74.49 | 59.96 | 22.85 | 8.34 | 81.55 | 62.86 | 19.14 | 16.81 |

**CIFAR-100** As for the classical setting (a), CIFAR-100 benchmarks train 200/400/800/1200 epochs from the stretch in Table A6, which is similar to CIFAR-10. Notice that we set weight decay to 0.0005 for cutting-based methods (CutMix, AttentiveMix+, SaliencyMix, FMix, ResizeMix) for better

Table A9: Top-1 accuracy (%) on Tiny based on ResNet (R) and ResNeXt-32x4d (RX). Notice that † denotes reproducing results with the official implementation, while other results are implemented with OpenMixup.

| Backbones | $\alpha$ | R-18 | RX-50 |
|---|---|---|---|
| Vanilla | - | 61.68 | 65.04 |
| MixUp | 1 | 63.86 | 66.36 |
| CutMix | 1 | 65.53 | 66.47 |
| ManifoldMix | 0.2 | 64.15 | 67.30 |
| SmoothMix | 0.2 | 66.65 | 69.65 |
| AttentiveMix+ | 2 | 64.85 | 67.42 |
| SaliencyMix | 1 | 64.60 | 66.55 |
| FMix | 1 | 63.47 | 65.08 |
| GridMix | 0.2 | 65.14 | 66.53 |
| ResizeMix | 1 | 63.74 | 65.87 |
| PuzzleMix | 1 | 65.81 | 67.83 |
| Co-Mixup† | 2 | 65.92 | 68.02 |
| AutoMix | 2 | 67.33 | 70.72 |
| SAMix | 2 | **68.89** | **72.18** |

performances when using ResNeXt-50 (32x4d) as the backbone. As shown in Table A7 using the modern setting (b), we train three modern architectures for 200/600 epochs from the stretch. We resize the raw images to $224 \times 224$ resolutions for DeiT-S and Swin-T, while modifying the stem network as the CIFAR version of ResNet for ConvNeXt-T with $32 \times 32$ resolutions. As shown in Table A8, we further provided more metrics to evaluate the robustness (top-1 accuracy on the corrupted version of CIFAR-100 (Hendrycks & Dietterich, 2019) and applying FGSM attack (Goodfellow et al., 2015)) and the prediction calibration.

**Tiny-ImageNet**    We largely follow the training setting of PuzzleMix (Kim et al., 2020) on Tiny-ImageNet, which adopts the basic augmentations of `RandomFlip` and `RandomResizedCrop` and optimize the models with a basic learning rate of 0.2 for 400 epochs with Cosine Scheduler. As shown in Table A9, all compared methods adopt ResNet-18 and ResNeXt-50 (32x4d) architectures training 400 epochs from the stretch on Tiny-ImageNet.

### B.3    Downstream Classification Benchmarks

We further provide benchmarks on three downstream classification scenarios in $224 \times 224$ resolutions with ResNet architectures, as shown in Table A10.

**Benchmarks on Fine-grained Scenarios.**    As for fine-grained scenarios, each class usually has limited samples and is only distinguishable in some particular regions. We conduct (a) transfer learning on CUB-200 and FGVC-Aircraft, and (b) fine-grained classification with training from scratch on iNat2017 and iNat2018. For (a), we use transfer learning settings on fine-grained datasets, using PyTorch official pre-trained models as initialization and training 200 epochs by SGD optimizer with the initial learning rate of 0.001, the weight decay of 0.0005, the batch size of 16, the same data augmentation as ImageNet-1K settings. For (b) and (c), we follow Pytorch-style ImageNet-1K settings mentioned above, training 100 epochs from the stretch.

**Benchmarks on Scenis Scenarios.**    As for scenic classification tasks, we study whether the mixup augmentations help the model to distinguish the backgrounds, which are less important than the foreground objects in commonly used datasets. We employ the PyTorch-style ImageNet-1K setting on Places205, which trains the model for 100 epochs with SGD optimizer, a basic learning rate of 0.1, and a batch size of 256.

### B.4    Transfer Learning

**Object Detection.**    We conduct transfer learning experiments with pre-trained ResNet-50 (He et al., 2016) and PVT-S (Wang et al., 2021) using mixup augmentations to object detection on COCO-2017 (Lin et al., 2014) dataset, which evaluate the generalization abilities of different mixup

Table A10: Top-1 accuracy (%) of mixup image classification with ResNet (R) and ResNeXt (RX) variants on fine-grained datasets (CUB-200, FGVC-Aircraft, iNat2017/2018) and Places205.

| Method | Beta $\alpha$ | CUB-200 R-18 | RX-50 | FGVC-Aircraft R-18 | RX-50 | Beta $\alpha$ | iNat2017 R-50 | RX-101 | iNat2018 R-50 | RX-101 | Beta $\alpha$ | Places205 R-18 | R-50 |
|---|---|---|---|---|---|---|---|---|---|---|---|---|---|
| Vanilla | - | 77.68 | 83.01 | 80.23 | 85.10 | - | 60.23 | 63.70 | 62.53 | 66.94 | - | 59.63 | 63.10 |
| MixUp | 0.2 | 78.39 | 84.58 | 79.52 | 85.18 | 0.2 | 61.22 | 66.27 | 62.69 | 67.56 | 0.2 | 59.33 | 63.01 |
| CutMix | 1 | 78.40 | 85.68 | 78.84 | 84.55 | 1 | 62.34 | 67.59 | 63.91 | 69.75 | 0.2 | 59.21 | 63.75 |
| ManifoldMix | 0.5 | 79.76 | 86.38 | 80.68 | 86.60 | 0.2 | 61.47 | 66.08 | 63.46 | 69.30 | 0.2 | 59.46 | 63.23 |
| SaliencyMix | 0.2 | 77.95 | 83.29 | 80.02 | 84.31 | 1 | 62.51 | 67.20 | 64.27 | 70.01 | 0.2 | 59.50 | 63.33 |
| FMix | 0.2 | 77.28 | 84.06 | 79.36 | 86.23 | 1 | 61.90 | 66.64 | 63.71 | 69.46 | 0.2 | 59.51 | 63.63 |
| ResizeMix | 1 | 78.50 | 84.77 | 78.10 | 84.0 | 1 | 62.29 | 66.82 | 64.12 | 69.30 | 1 | 59.66 | 63.88 |
| PuzzleMix | 1 | 78.63 | 84.51 | 80.76 | 86.23 | 1 | 62.66 | 67.72 | 64.36 | 70.12 | 1 | 59.62 | 63.91 |
| AutoMix | 2 | 79.87 | 86.56 | 81.37 | 86.72 | 2 | 63.08 | 68.03 | 64.73 | 70.49 | 2 | 59.74 | 64.06 |
| SAMix | 2 | **81.11** | **86.83** | **82.15** | **86.80** | 2 | **63.32** | **68.26** | **64.84** | **70.54** | 2 | **59.86** | 64.27 |

Table A11: Trasfer learning of object detection task with ImageNet-1k pre-trained ResNet-50 on COCO.

| Method | IN-1K Acc | COCO mAP | $AP_{50}^{bb}$ | $AP_{75}^{bb}$ |
|---|---|---|---|---|
| Vanilla | 76.8 | 38.1 | 59.1 | 41.8 |
| Mixup | 77.1 | 37.9 | 59.0 | 41.7 |
| CutMix | 77.2 | 38.2 | 59.3 | 42.0 |
| ResizeMix | 77.4 | 38.4 | 59.4 | 42.1 |
| PuzzleMix | 77.5 | 38.3 | 59.3 | 42.1 |
| AutoMix | 77.9 | 38.6 | 59.5 | **42.2** |
| SAMix | **78.1** | **38.7** | **59.6** | **42.2** |

Table A12: Trasfer learning of object detection task with Mask R-CNN and semantic segmentation with Semantic FPN with pre-trained PVT-S on COCO and ADE20K.

| Method | IN-1K Acc | COCO mAP | $AP_{50}^{bb}$ | $AP_{75}^{bb}$ | ADE20K mIoU |
|---|---|---|---|---|---|
| MixUp+CutMix | 79.8 | 40.4 | 62.9 | 43.8 | 41.9 |
| AutoMix | 80.7 | 40.9 | 63.9 | 44.1 | 42.5 |
| TransMix | 80.5 | 40.9 | 63.8 | 44.0 | 42.6 |
| TokenMix | 80.6 | **41.0** | **64.0** | 44.3 | **42.7** |
| TokenMixup | 80.5 | 40.7 | 63.6 | 43.9 | 42.5 |
| SMMix | **81.0** | **41.0** | 63.9 | **44.4** | **43.0** |

approaches. We first fine-tune Faster RCNN (Ren et al., 2015) with ResNet-50-C4 using Detectron2 (Wu et al., 2019) in Table A11, which is trained by SGD optimizer and multi-step scheduler for 24 epochs (2×). The *dynamic* mixup methods (*e.g.,* AutoMix) usually achieve both competitive performances in classification and object detection tasks. Then, we fine-tune Mask R-CNN (He et al., 2017) by AdamW optimizer for 24 epochs using MMDetection (Chen et al., 2019) in Table A12. We have integrated Detectron2 and MMDetection into OpenMixup, and the users can perform the transferring experiments with pre-trained models and config files. Compared to *dynamic* sample mixing methods, recently-proposed label mixing policies (*e.g.,* TokenMix and SMMix) yield better performances with less extra training overheads.

**Semantic Segmentation.** We also perform transfer learning to semantic segmentation on ADE20K (Zhou et al., 2018) with Semantic FPN (Kirillov et al., 2019) to evaluate the generalization abilities to fine-grained prediction tasks. Following PVT (Wang et al., 2021), we fine-tuned Semantic FPN for 80K interactions by AdamW (Loshchilov & Hutter, 2019) optimizer with the learning rate of $2 \times 10^{-4}$ and a batch size of 16 on $512^2$ resolutions using MMSegmentation (Contributors, 2020b). Table A12 shows the results of transfer experiments based on PVT-S.

### B.5 RULES FOR COUNTING THE MIXUP RANKINGS

We have summarized and analyzed a great number of mixup benchmarking results to compare and rank all the included mixup methods in terms of *performance*, *applicability*, and the *overall* capacity. Specifically, regarding the *performance*, we averaged the accuracy rankings of all mixup algorithms for each downstream task, as well as averaging their robustness and calibration results rankings separately. Finally, these ranking results are averaged again to produce a comprehensive range of performance ranking results. As for the *applicability*, we adopt a similar ranking computation scheme considering the *time usage* and the *generalizability* of the methods. With the *overall* capacity ranking, we combined the performance and applicability rankings with a 1:1 weighting to obtain the final take-home rankings. For equivalent results, we take a tied ranking approach. For instance, if there are three methods tied for first place, then the method that actually results in fourth place is recorded as second place by default. Finally, we provide the comprehensive rankings as shown in Table 5.

