# OpenReview forum: "OpenMixup: A Comprehensive Mixup Benchmark for Visual Classification"
_ICLR.cc/2024/Conference — Submitted to ICLR 2024_

### Official Review · Reviewer_nCz4 · 2023-10-30

**Soundness:** 3 good
**Presentation:** 3 good
**Contribution:** 2 fair
**Rating:** 5
**Confidence:** 4

**Summary:**

This paper introduces a comprehensive benchmark for image classification methods based on *mixup*. The benchmark is constructed from a unified codebase and offers a detailed overview of the codebase itself, the implemented methods, available datasets, and the evaluation metrics available. Moreover, the paper includes an empirical analysis that compares the performance (accuracy) of various models across diverse datasets, as well as a comparison of the trade-off between performance, total training time and memory usage.

**Strengths:**

- **Comprehensive Analysis with practical relevance**: The paper presents an extensive examination of various mixup methods for image classification. It assesses these methods across a wide range of classification datasets and employs multiple architectures. This comprehensive approach ensures that the study's findings are robust and applicable to various real-world scenarios. By considering the trade-off between performance, training time, and GPU memory usage, it provides valuable insights for implementing mixup techniques.
- **Open-Source Codebase**: Open-source code promotes transparency and reproducibility in research. This contributes to the credibility and reliability of further study on mixup methods.

**Weaknesses:**

- **Lack of clear Take-Home Message**: The paper is a tech report that describes the functionalities and the different components in the library. While I do not dismiss the massive work put behind building the library, the paper is missing a clear take-home message from the "interesting observations" claimed by the authors. There is no substantial findings or actionable insights beyond the existence of the associated codebase.
- **Misalignment of Evaluation Metrics**: The paper introduces several evaluation metrics which are not applied in the experiments, such as calibration or robustness. The primary focus remains on accuracy, which as already been extensively studied in the papers associated with each individual method. This limits the paper's potential to offer a fresh perspective on the methods by not exploring alternative criteria beyond accuracy, especially as other evaluation metrics are already described as available in the benchmark.

### Remarks:

- The color gradient in the plots make it difficult to really differentiate two methods.
- The double-blind nature of the review is broken by keeping the name of the contributors and authors of the associated arxiv paper in the code provided as supplementary material ...

**Questions:**

- What would be the main take-home message from the empirical study presented ? Beyond the existence of the open-source library, what should we remember from this paper ?
- I really think that the paper would be improved with a comparison of the methods on other evaluation metrics than accuracy, it might provide a different perspective on the implemented methods. Though I totally understand that this would require a lot of new experiments.

---

> ### Author Response · Authors · 2023-11-22
> **Response to Reviewer nCz4**
>
> We would like to extend our gratitude for raising the constructive points to help us improve the manuscript. Based on your questions and suggestions, we have presented our revision and general response. All revised information in our revision is highlighted for your convenience.
>
> We would like to address your concerns and the points you raised for improvement:
>
> ### **1. On the Clear Take-Home Message.**
>
> Thank you for your valuable comments. Based on your advice, we have summarized and analyzed a great number of benchmarking results to compare and rank all the included mixup methods in terms of **performance, applicability,** and **overall capability**.
>
> Specifically, regarding performance, we averaged the accuracy rankings of all mixup algorithms for each downstream task, as well as averaging their robustness and calibration (supplementary experiments) rankings separately. Finally, these ranking averages were averaged again to produce the comprehensive range of performance ranking results. As for speed, we adopt a similar ranking computation scheme. With the overall capability ranking, we combined the performance and applicability rankings with a 1:1 weighting to obtain the final take-home rankings.
>
> Finally, we provide the **take-home rankings** as shown in the following tables. Again, we appreciate your constructive suggestion.
>
>
> |               | Mixup | CutMix | DeiT | SmoothMix | GridMix | ResizeMix | ManifoldMix | FMix | AttentiveMix | SaliencyMix | PuzzleMix | AlignMix | AutoMix | SAMix | TransMix | SMMix |
> |---------------|:-----:|:------:|:----:|:---------:|:-------:|:---------:|:-----------:|:----:|:------------:|:-----------:|:---------:|:--------:|:-------:|:-----:|:--------:|:-----:|
> | Performance   |   13  |   11   |   5  |     16    |    15   |     8     |      12     |  14  |       7      |      9      |     6     |    10    |    2    |   1   |     4    |   3   |
> | Applicability |   1   |    1   |   1  |     1     |    1    |     1     |      1      |   1  |       3      |      1      |     4     |     2    |    7    |   6   |     5    |   5   |
> | Overall       |   8   |    6   |   1  |     11    |    10   |     4     |      7      |   9  |       5      |      5      |     5     |     6    |    4    |   2   |     4    |   3   |
>
>
> ### **2. Misalignment of Evaluation Metrics.**
>
> Thank you for raising the issue we had with evaluation metrics. Based on your suggestion, we have revised the article where the presentation was not rigorous. Meanwhile, for robustness and calibration evaluation, we have added the related experiments, as shown in the table below.
>
> The results show that although these new evaluation metrics provide a variety of interpretable perspectives, the conclusions they provide are still consistent with those of accuracy. This is also consistent with what we know about mixup as an augmentation method. An augmentation method is good as long as it is effective, fast, and can be applied to multiple scenarios concurrently. Furthermore, we have provided comprehensive radar plots in the paper, which clearly reflect the performance of each algorithm in different scenarios. We suppose this is a comprehensive evaluation of existing mixup algorithms. Nonetheless, we have added the corresponding experiments you mentioned and have come up with consensus results.
>
> ### **3. The Color Gradient in the Plots.**
>
> We appreciate that you pointed out the lack of differentiation of color gradients. Since we cover a lot of algorithms, selecting various color gradients can be more or less difficult to differentiate. We have tried to make changes in our revision to make the color gradients in plots more distinguishable. Again, thank you for your valuable suggestions.
>
> ### **4. Comparison with Other Evaluation Metrics.**
>
> As mentioned in the second response, based on your suggestion, we have added the related robustness and calibration evaluation experiments, as shown in Table A8 in the revised version. The results show that although these new evaluation metrics provide a variety of interpretable perspectives, the conclusions they provide are still consistent with those of accuracy. This is also consistent with what we know about mixup as an augmentation method. An augmentation method is good as long as it is effective, fast, and can be applied to multiple scenarios concurrently. Furthermore, we have provided comprehensive radar plots in the paper, which clearly reflect the performance of each algorithm in different scenarios. We suppose this is a comprehensive evaluation of existing mixup algorithms. However, we have added the corresponding experiments you mentioned and have come up with consensus results.
>
> Overall, we sincerely appreciate all your time and valuable feedback. We hope that you will reconsider our submission with the revision and response. If you are satisfied with our response, please consider updating your score. If you need any clarification, please feel free to contact us.

---

### Official Review · Reviewer_x5Ek · 2023-10-31

**Soundness:** 2 fair
**Presentation:** 3 good
**Contribution:** 2 fair
**Rating:** 5
**Confidence:** 5

**Summary:**

This paper presents the mixup visual classification benchmark, called OpenMixup, which also offers a unified modular mixup-based model design and training codebase framework for visual representation learning. Extensive experiments on both static and dynamic mixup baselines are conducted across various visual classification datasets.

**Strengths:**

1. The paper is well-organized and easy to follow.
2. The experimental assessment is comprehensive.

**Weaknesses:**

1. This article is a summary of the previous Mixup-based data augmentation approaches and does not propose new tasks and methods.

2. Lack of important references: "Zhao Q, Huang Y, Hu W, et al. MixPro: Data Augmentation with MaskMix and Progressive Attention Labeling for Vision Transformer[C]//The Eleventh International Conference on Learning Representations. 2022."

3. Lack of latest ViT's backbone results for benchmarks.
Such as FastViT: "Vasu P K A, Gabriel J, Zhu J, et al. FastViT: A Fast Hybrid Vision Transformer using Structural Reparameterization[J]. //Proceedings of the IEEE/CVF International Conference on Computer Vision (2023)"

**Questions:**

None.

---

> ### Author Response · Authors · 2023-11-22
> **Response to Reviewer x5Ek**
>
> We sincerely appreciate all the time you took to thoroughly review our work and your valuable feedback. We are delighted about your assessment of our paper as a well-organized and comprehensive assessment.
>
> Again, we would like to express our appreciation to you for pointing out the issue of academic advancements (new tasks and methods). Regarding the points you raised for improvement:
>
> ### **1. No New Tasks and Methods.**
>
> First, we have elaborated on **why we need a comprehensive mixup benchmark.** and **why we need a standardized open-source platform.** in the abstract and Sec.1. Briefly, mixup is a widely-used data augmentation method in modern Convolutional Neural Networks (CNNs) and Visual Transformers (ViTs). To the best of our knowledge, however, as a fundamental building block of today's computer vision architecture, there has been no such comprehensive benchmarking study (which is both **time-consuming** and **computationally resource-intensive**) for the mixup community to drive progress in a standard manner. As researchers, we believe that we can all agree that an area will **struggle to advance without a comprehensive benchmark**. This is the first reason that we focus on mixup, especially for building a comprehensive mixup visual classification benchmark. **It is quite urgent work for the community**. **This area needs a benchmark.** In this context, our work is indeed **the first** mixup visual classification benchmark. Even though we do not present new algorithms or define new tasks, we are still highly meaningful to the community as the first benchmarking study and a unified codebase platform.
>
> Second, we have to emphasize that most current mixup methods are crafted with diverse angles of enhancement, applying distinct configurations and even different coding styles **without an all-in-one open-source coding platform** for streamlined pre-processing, method design, training, and evaluation. The lack of standardization **poses obstacles to fair comparison and effortless development**, necessitating **costly trial-and-error for researchers**. This is the second reason that we focus on mixup augmentations, especially for providing a unified model design and training platform.
>
> In summary, we conclude our main contributions and values as follows:
>
> **(a)** Mixup is an augmentation method that is widely employed nowadays. However, there is no comprehensive benchmark for mixup algorithms so far. This will lead to the **absence of a standardized criterion** for numerous mixup classification methods, and researchers have no idea how to choose a proper method. Personally, there are many researchers who reached out to me and stated that the existing mixup classification algorithms are **varied and full of pitfalls**.
>
> To address this issue, OpenMixup is presented as **the first** comprehensive mixup visual classification benchmark. It first enables researchers to **fairly compare** the performance of their models against a broad set of mixup baselines, providing a **clear and objective measurement** of the method's effectiveness. Such a first-of-its benchmarking study is both **time-consuming** and **resource-intensive**, but we still carried it out **from scratch**.
>
> I suppose this makes our work truly meaningful and full of value, even though the scope is objectively limited.
>
> **(b)** Apart from the benchmarking study, we provide **a standardized model design and training codebase platform** for mixup-based visual classification. The overall framework is streamlined and modular, which can enhance the **accessibility and customizability of downstream deployment and applications**.
>
> **(c) Interesting observations, insights, and take-home conclusive rankings** on comprehensive performance, robustness, training stability, and convergence are obtained (**In Sec.4**) with diverse analysis toolkits (*e.g.* **accuracy radar plots**, **trade-off plots**, loss landscape, CAM visualization, *etc.*) from our OpenMixup platform, enabling researchers to identify what specific properties contribute to the success of different methods. In the revision, we aim to provide take-home ranking results that offer a clear message, helping researchers select the proper mixup methods. Archers to identify what specific properties contribute to the success of different methods.
>
> ### **2. The Important Reference.**
>
> Thanks for pointing out the missing reference. we have added this important work in the revision. Thanks again.
>
> ### **3. The FastViT Backbone Results.**
>
> We appreciate you pointing out our missing backbone. Due to the limited time and computational resources we have in the rebuttal period, we would add this work to the reference first and conduct related experiments on this work in our subsequent revisions. Thanks again.
>
> In conclusion, we sincerely appreciate all your valuable feedback to help us improve our work. We hope that you will reconsider our submission with the revised version.

---

### Official Review · Reviewer_kQcu · 2023-11-01

**Soundness:** 3 good
**Presentation:** 3 good
**Contribution:** 2 fair
**Rating:** 5
**Confidence:** 5

**Summary:**

In this paper, the authors proposed OpenMixup, a mixup based benchmark for visual classification. Multiple algorithms are implemented in a unified framework and evaluated on a wide spectrum of classification datasets.

**Strengths:**

1. It is very helpful to have a unified and comprehensive benchmark for mixup algorithms given that it has been widely used in model training.
2. Extensive implementation and evaluation are conducted for mixup based models. The authors also provide good visualizations and analysis on the results.
3. Writing is good and easy to follow.

**Weaknesses:**

1. The main contribution seems to be the implementation of mixup based models. I am not sure ICLR is a good venue for this kind of work.
2. Detection and Segmentation are currently implemented as Transfer Learning from classification trained backbones which is quite limited as there are also mixup models for detection.

**Questions:**

1. How this engineering focused and narrowed application benchmark benefit the wide community of ICLR?
2. How do you extend the current framework to add the mixup algorithms for object detection that are not based on transfer learning?

---

> ### Author Response · Authors · 2023-11-22
> **Response to Reviewer kQcu**
>
> We first express our gratitude for your time and effort in reviewing our submission. We have made numerous adjustments to our submission and provide a comprehensive general response. We invite you to first go through our general responses because they may answer most of your questions.
>
> Again, we would like to express our appreciation to you for pointing out the issue of the scope and the value to the community. Here, we will address your concern from the following two aspects:
>
> ### **1. On the Value and Contributions of OpenMixup.**
>
> We appreciate your valuable comments. We present a comprehensive overview of our contributions and novelties to the community.
>
> **(1)** Mixup is an augmentation method that is widely employed nowadays. However, there is no comprehensive benchmark for mixup so far. This will lead to the **absence of a standardized criterion** for numerous mixup classification methods, and researchers have no idea how to choose a proper method. Personally, there are many researchers who reached out to me and stated that the existing mixup classification algorithms are **varied and full of pitfalls**. As researchers, we believe that we can all agree that an area will **struggle to advance without a comprehensive benchmark**. It is quite urgent for the community. This area needs a benchmark.
>
> To address this issue, OpenMixup is presented as **the first** comprehensive mixup visual classification benchmark. It first enables researchers to **fairly compare** the performance of their models against a broad set of mixup baselines, providing a **clear and objective measurement** of the method's effectiveness. Such a first-of-its benchmarking study is both **time-consuming** and **resource-intensive**, but we still carried it out **from scratch**.
>
> I suppose this makes our work truly meaningful and full of value, even though the scope is objectively limited.
>
> **(2)** Apart from the benchmarking study, we provide **a standardized model design and training codebase platform** for mixup-based visual classification. The overall framework is streamlined and modular, which can enhance the **accessibility and customizability of downstream deployment and applications**.
>
> **(3)** **Interesting observations and insights** on comprehensive performance, robustness, training stability, and convergence are obtained (**In Sec.4**) with diverse analysis toolkits (*e.g.* **accuracy radar plots**, **trade-off plots**, loss landscape, CAM visualization, *etc.*) from our OpenMixup platform, enabling researchers to identify what specific properties contribute to the success of different methods. In the revision, we aim to provide take-home ranking results that offer a clear message, helping researchers select the proper mixup methods.
>
> ### **2. Without Transfer Learning Detection and Segmentation Tasks.**
>
> We appreciate your valuable comment. First, as emphasized in our revision (**in Sec.1**), although mixup exerts its power in other downstream tasks like object detection, it is derived from its supervised classification counterpart with **shared attributes**. Meanwhile, mixup augmentations utilized in segmentation tasks need to be analyzed and specially designed on a case-by-case basis since most of the mixup methods for classification tasks cannot be directly used in segmentation tasks. This is less relevant to the study of the properties of the mixup algorithm itself. We suppose that the study of mixup classification is sufficient to provide a comprehensive understanding of mixup.
>
> Second, while our main focus is indeed on supervised visual classification tasks with mixup augmentations, the platform we provided, OpenMixup, can be extended to other tasks, as you mentioned, such as object detection and semantic segmentation. We have provided **Transfer Learning to Semantic Segmentation** and more detailed benchmarking results in the revised supplementary materials to better evaluate the mixup methods. We believe that the unified streamlined platform we provided could be beneficial for these downstream tasks, and we intend to extend our work in these directions in the future.
>
> In conclusion, we appreciate all your constructive feedback and the opportunity given to improve our OpenMixup. We believe that addressing these points has significantly enhanced our paper and its contributions to the mixup community. We hope that you will reconsider our submission with the revision and response. If you are satisfied with our response and effort, please consider updating your score. If you need any clarification, please feel free to contact us.

---

> > ### Comment · Reviewer_kQcu · 2023-11-22
> >
> > Thanks for the feedback. I am not convinced that "time-consuming", "resource-intensive", and "from scratch" would be good evidence for value to ICLR. I think there is value of this work but as mentioned in my initial comments "I am not sure ICLR is a good venue for this kind of work." I am sticking to my initial rating.

---

> > > ### Author Response · Authors · 2023-11-22
> > >
> > > Thanks for your timely reply and suggestions. We have also provided useful conclusions in Sec. 4 (e.g., the take-home message in Table 5) and believe these will benefit the community in both academic research and practical applications of mixup augmentations. Once again, we express our sincere appreciation for your time and efforts in reviewing our work.

---

### Official Review · Reviewer_rFDa · 2023-11-01

**Soundness:** 3 good
**Presentation:** 3 good
**Contribution:** 3 good
**Rating:** 6
**Confidence:** 4

**Summary:**

The authors present the first benchmark framework which employ various published mixup methods for visual classification tasks. Not only on classification dataset, they also publish the downstream task baselines which mixup researches usually evaluate their methods as a strong and effective regularizer. To evaluate mixup researches on various tasks, they provide visualization tools, various dataset module, and various models which are widely used on research.

**Strengths:**

1.	This work is well-motivated. Since the programs used in the studies of the mixup methodologies differed, it was difficult to compare them to each other. This benchmark framework is useful to the mixup research community and practical field.
2.	The diversity of images and learning models generated by the proposed method, as well as the diversity of downstream tasks, can provide a complete picture of the performance of the training intervals and, in particular, reveal weaknesses.

**Weaknesses:**

1.	For the installation of OpenMixup, detailed settings are required, such as the version that the authors tested. Because the framework requires pytorch-based frameworks ‘mmcv’, it requires more specifics like author-tested versions of pytorch, mmcv, and mmclassification. Including other tasks, such as detection and segmentation, the description of specific versions would make it more accessible to other researchers.
2.	Extensibility to add new mixup method. As the authors mentioned, we hope this benchmark can be extended to other mixup methods and scenarios not only in research fields but also in practical applications. To this end, instructions to add the mixup method could be included in the document. Recent mixup methods include additional modules (e.g. AutoMix, SAMix, RecursiveMix, SMMix,. Etc); therefore, the guideline for the methodologies that require additional modules would be better to be included.
3.	To measure the robustness metric using OpenMixup, the performances on the corrupted dataset (Cifar-100-C or ImageNet-C) could be included.

**Questions:**

1.	In the latest version, we cannot find TransMix’s implementation of Swin-tiny, while the authors describe it as the supported method in OpenMixup. Then, how can the authors get TransMix score of Deit-T on ImageNet-1k in Table A3, A7? (including SMMix)
2.	Why does CutMix's hyper-parameter alpha vary from experiment to experiment, and why doesn't it utilize the officially utilized value of 1? Is there a reason for writing it differently for different methods?
3.	As far as I know, the official code of OpenMixup provides training segmentation tasks using pre-trained backbone model. Why do not the authors include the results?

---

### Author Response · Authors · 2023-11-22
**Official Comment by Authors**

Dear Reviewers,

Greetings!

We would like to first sincerely express our appreciation for the time and effort all 4 reviewers dedicated to reviewing our submission. We find all your comments to be exceptionally helpful in making improvements to our submission. Based on your feedback, we have individually responded to each reviewer's comments in terms of the **value and benefits for both the research and engineering community**. Moreover, problems regarding the experiment setup and results, as well as presentation clarity mentioned by some of the reviewers, have been addressed and improved in the revision. The key points are highlighted in magenta color. With the help of the reviewers, we have more accurately illustrated **the problems we addressed**, **the contributions and innovations we made**, and **the empirical conclusions we drew** in our revision.

We have noticed that there seem to be specific questions in common. In order to give you a better illustration of the value of our work and a picture of the changes we have made in our revision, we would like to provide further clarification as follows:

### 1. Contributions to the Community

Several reviewers expressed concerns about the research novelty and benefits of OpenMixup to the ICLR community. We have a deep appreciation for the reviewers' comments. We have emphasized the reason why this benchmarking research is of significance ****in the responses.  Here, we present a comprehensive overview of our contributions and novelties, taking into account the reviewers’ comments and the improvements made in our revision:

**(1)** OpenMixup is the **first** comprehensive mixup visual classification benchmark. It enables researchers to **fairly compare** the performance of their models against a broad set of mixup baselines, providing a **clear and objective measurement** of the method's effectiveness. As a fundamental building block of today's computer vision architecture, there has been no such comprehensive benchmarking study for the mixup community. As researchers, I believe we can all agree that an area will struggle to move forward without a comprehensive benchmark. More importantly, such a first-of-its benchmarking study is both **time-consuming** and **resource-intensive**, but we still carried it out **from scratch** for the community.

**(2)** We provide **a standardized model design and training codebase platform** for mixup-based visual classification. The overall framework is streamlined and modular, which can enhance the **accessibility and customizability of downstream deployment and applications**.

**(3)** Empirical analysis is conducted on various scenarios. **Interesting observations, insights and take-home conclusion rankings** are obtained (Please refer to **Sec.4**) with diverse analysis toolkits (*e.g.* **accuracy radar plots**, **trade-off plots**, loss landscape, CAM visualization, *etc.*) from our OpenMixup platform, enabling researchers to identify what specific properties contribute to the success of different methods.

### 2. Evaluation Metrics

Based on the reviewers’ suggestion, we have added the robustness and calibration evaluation experiments to enrich our evolutions. The results show that although these new evaluation metrics provide a variety of interpretable perspectives, the conclusions they provide are still consistent with those of accuracy. This is also consistent with what we know about mixup as an augmentation method. An augmentation method is good as long as it is effective, fast, and can be applied to multiple scenarios concurrently. Furthermore, we have provided comprehensive radar plots in the paper, which clearly reflect the performance of each algorithm in different scenarios. We suppose this is a comprehensive evaluation of existing mixup algorithms. However, we have added the corresponding experiments you mentioned and have come up with consensus results.

In conclusion, we appreciate all your time and valuable feedback. Our paper introduces **the first** comprehensive mixup benchmarking study and a unified practical codebase platform for the mixup community. We hope that you will reconsider our submission with the revision and response. If you are satisfied with our response and effort, please consider updating your score. If you need any clarification, please feel free to contact us.

Best regards,

Authors

---

### Author Response · Authors · 2023-11-23
**Official Comment by Authors**

Dear Reviewers,

We greatly appreciate your effort and valuable feedback. In our response, we carefully illustrated and answered all your questions in detail. Additional experiments and analysis results are also provided as you requested. In addition, we further clarified several unclear statements in the paper. We have incorporated all changes into the revised manuscript for your consideration. We hope your concerns have been addressed.

As you may know, unlike previous years, the discussion period this year can only last until November 22, and we are gradually approaching this deadline. We take it seriously and would like to discuss this with you during this time. We would be happy to provide more information based on your feedback or further questions.

If you are satisfied with our response, please consider updating your score. If you need any clarification, please feel free to contact us. We would eagerly welcome any further guidance at your convenience!

Best regards,

Authors

---

### Meta-Review · Area_Chair_TcUE · 2023-12-22

**Metareview:**

The paper presents OpenMixup, the first comprehensive benchmarking study for supervised visual classification, focusing on the data-dependent augmentation technique of mixup. OpenMixup includes a unified mixup-based model design and training framework, provides extensive evaluations across 12 image datasets and insights into the impact of mixup policies, network architectures, and dataset properties on visual classification performance.

While the reviewers acknowledged the importance of the benchmark, they have raised several important concerns: 1) lack of technical contribution and substantial findings – see all reviewer comments, 2) mixup models for detection beyond transfer learning baselines – see Reviewer kQcu comments, 3) lack of references and latest baseline/backbone  - see Reviewer x5Ek comments, 4) misalignment of evaluation metrics – see Reviewer nCz4 comment.

The rebuttal was able to clarify some questions, but did not manage to sway any of the reviewers. A general consensus among reviewers and AC was reached to reject the paper. We hope the reviews are useful for improving and revising the paper.

**Justification For Why Not Higher Score:**

AC is in consensus with three out of four reviewers.

**Justification For Why Not Lower Score:**

N/A

---

### Decision · Program_Chairs · 2024-01-16

Reject